META-RESEARCH ARTICLE

# Educating the future generation of researchers: A cross-disciplinary survey of trends in analysis methods

**Taylor Bolt[1]\*, Jason S. Nomi[1], Danilo Bzdok[2,3], Lucina Q. Uddin[1,4]**

**1** Department of Psychology, University of Miami, Coral Gables, Florida, United States of America,
**2** Department of Biomedical Engineering, McConnell Brain Imaging Centre (BIC), Montreal Neurological Institute (MNI), Faculty of Medicine, McGill University, Montreal, Canada, **3** Mila—Quebec Artificial Intelligence Institute, Montreal, Canada, **4** Neuroscience Program, University of Miami Miller School of Medicine, Miami, Florida, United States of America

\* tsb46@miami.edu

**Data Availability Statement:** Two sources of peer-reviewed literature were used for this analysis: the Pubmed Central Open Access Subset (PMC OAS) (N = 2,869,889 articles at time of study); and the Pubmed Central Author Manuscript (PMC AM)

## Abstract

Methods for data analysis in the biomedical, life, and social (BLS) sciences are developing at a rapid pace. At the same time, there is increasing concern that education in quantitative methods is failing to adequately prepare students for contemporary research. These trends have led to calls for educational reform to undergraduate and graduate quantitative research method curricula. We argue that such reform should be based on data-driven insights into within- and cross-disciplinary use of analytic methods. Our survey of peer-reviewed literature analyzed approximately 1.3 million openly available research articles to monitor the cross-disciplinary mentions of analytic methods in the past decade. We applied data-driven text mining analyses to the "Methods" and "Results" sections of a large subset of this corpus to identify trends in analytic method mentions shared across disciplines, as well as those unique to each discipline. We found that the *t* test, analysis of variance (ANOVA), linear regression, chi-squared test, and other classical statistical methods have been and remain the most mentioned analytic methods in biomedical, life science, and social science research articles. However, mentions of these methods have declined as a percentage of the published literature between 2009 and 2020. On the other hand, multivariate statistical and machine learning approaches, such as artificial neural networks (ANNs), have seen a significant increase in the total share of scientific publications. We also found unique groupings of analytic methods associated with each BLS science discipline, such as the use of structural equation modeling (SEM) in psychology, survival models in oncology, and manifold learning in ecology. We discuss the implications of these findings for education in statistics and research methods, as well as within- and cross-disciplinary collaboration.

## Introduction

The methodological landscape of the biomedical, life, and social (BLS) sciences is becoming increasingly complex. This increasing complexity is driven by the advent of open-source

collection (N = 659,133 articles). The PMC OAS provides access to full-texts from a total of 14,722 open access peer-reviewed journals (at time of study). The PMC AM collection provides access to full texts of manuscripts made available in PMC by authors in compliance with the NIH Public Access Policy. Both sources form part of PMC's open access collection (37) (https://www.ncbi.nlm.nih.gov/pmc/tools/textmining/). Bulk downloads of the full OAS and AM collection articles were conducted using the PMC FTP service.

**Funding:** This work was supported by grants from the Canadian Institute for Advanced Research (DB), a Gabelli Senior Scholar Award from the University of Miami (LQU), a grant from the Social Science Research Council (LQU), an R01MH107549 from the National Institute of Mental Health (NIMH) (LQU), an R03MH121668 from the Institute of Mental Health (NIMH) (JSN), and a NARSAD Young Investigator Award to (JSN). The funders had no role in study design, data collection and analysis, decision to publish, or preparation of the manuscript.

**Competing interests:** The authors have declared that no competing interests exist.

**Abbreviations:** ANN, artficial neural network; ANOVA, analysis of variance; BLS, biomedical, life, and social; CNN, convolutional neural network; GEE, generalized estimating equation; GLIM, generalized linear model; ICA, independent component analysis; MDS, multidimensional scaling; MNB, multinomial naive Bayes; NER, named entity recognition; NMF, nonnegative matrix factorization; PCA, principal component analysis; PLS/DA, partial least squares/discriminant analysis; PMC AM, PubMed Central Author Manuscript; PMC OAS, PubMed Central Open Access Subset; ROC, receiver operating characteristic; SEM, structural equation modeling.

science [1], the availability of large, complex datasets [2–4], and increasing computational resources [5–7]. The classic statistical tools (e.g., *t* test, analysis of variance (ANOVA), and linear regression) taught in introductory statistics courses are, at times, perceived insufficient to prepare researchers for the age of big data, machine learning, and open-source software. Concerned that BLS sciences educational training is struggling to keep up with these trends, many researchers and statisticians have advocated for educational reform to introductory research methods and statistics courses [8–13]. We argue that a crucial step in this direction is a more complete understanding of actual trends in analytic method usage across BLS sciences. Such an understanding will offer valuable insights into the necessary methodological skills and knowledge needed to train early career scientists for future success in their disciplines and their interdisciplinary collaborations.

Increasingly, analytic methods developed in one discipline find fruitful application in another. For example, deep learning, a machine learning technique developed by the artificial intelligence community, has successfully been used by biologists to predict three-dimensional protein structure [14]. The explosive adoption of neural networks across biology and multiple other fields illustrates the need for educational training to note these trends and keep pace with the demand for expertise in these emerging advanced analytic approaches.

In this study, we conducted a systematic charting of analytic method usage across BLS disciplines over time. We applied natural language processing tools to a large corpus of open-access peer-reviewed literature. Our study aimed to map out the methodological landscape of the BLS disciplines and identify changing trends over the past decade (2009 to 2020). Here, we use the term "analytic methods" to broadly denote any quantitative or qualitative method for data analysis, including any algorithms, statistics, or models used to describe, summarize, or interpret a sample of data. This definition is meant to exclude those elements of research methodology involved in data collection or experimental or study design. "Study" is also broadly defined as a peer-reviewed quantitative- or qualitative-based assessment of measured data points, including experimental, observational, or meta-analytic research. We retraced trends in analytic methods across (1) time; and (2) research disciplines. From a temporal perspective, we identified analytic methods that have increased or decreased in prominence across BLS disciplines over the past decade (2009 to 2020). From a cross-disciplinary perspective, we identified analytic methods that are uniquely prominent within each BLS discipline and the similarity or dissimilarity of BLS disciplines, in terms of their usage of analytic methods.

Our survey found that analytic methods commonly taught in introductory research methods and statistics courses (e.g., *t* test, ANOVA) remain the most commonly mentioned methods in BLS research articles over the past decade. However, these methods have largely declined in prominence, or remained stable, over the past decade. On the other hand, multivariate statistics and machine learning methods have exhibited a consistent, sometimes exponential, increase in mentions from 2009 to 2020. Further, we found that analytic methods are not equally distributed across BLS disciplines, but tend to cluster into certain disciplines over others.

## Results

### Preprocessing and analysis summary

The primary goal of this study is to describe and understand usage shifts in analytic methods across BLS disciplines over time. We analyzed approximately 1.3 million articles published in a decade of research to accomplish this goal. We extracted mentions/adoptions of analytic methods from "Methods and materials" and "Results" sections of a large corpus of peer-reviewed articles (PubMed Central Open Access Subset, PMC OAS [15]). We used a named

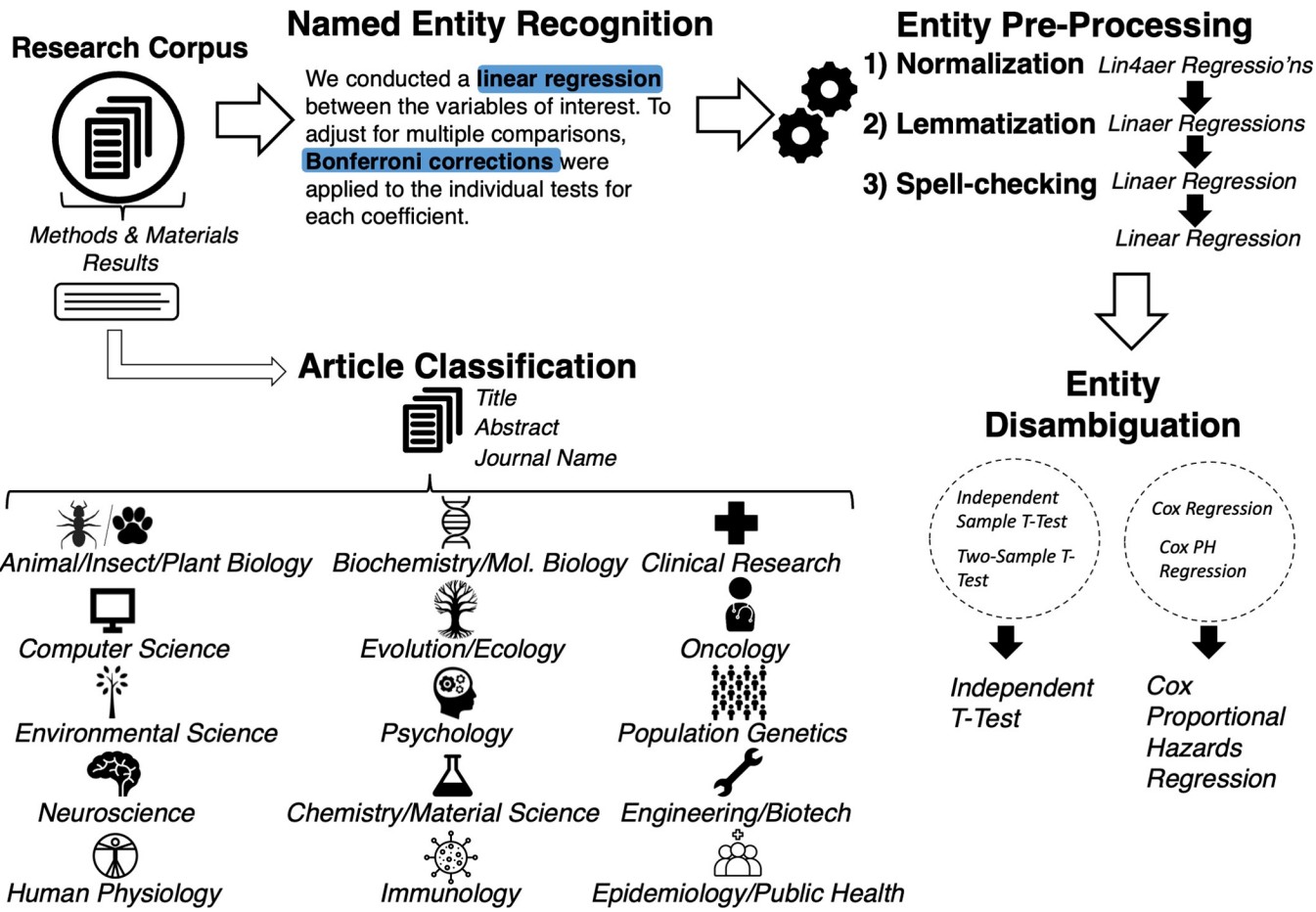

**Fig 1. Preprocessing pipeline.** (1) Retrieval and parsing of full-text "Methods and materials" and "Results" sections, (2) article classification into BLS disciplines, (3) NER of analytic method entities, (4) entity string preprocessing, and (5) a manual entity disambiguation step whereby analytic method entities are disambiguated into equivalent analytic methods (e.g., independent sample *t* test, and Cox proportional hazards regression). BLS, biomedical, life, and social; NER, named entity recognition.

entity recognition (NER) algorithm trained specifically for this purpose. We refer to these extracted mentions from the text as analytic method entities—unique strings of alphanumeric characters that refer to a distinct method for data analysis. The extracted entities then underwent a sequence of preprocessing steps including removal of unwanted characters and lemmatization (i.e., removing inflectional endings). The preprocessing workflow included a manual entity disambiguation step that classified entities referring to equivalent analytic methods to the same category—e.g., "Cox regression" and "Cox PH regression" were both classified as "Cox proportional hazards regression." The final number of unique analytic method entities after these preprocessing steps was $N = 250$. In addition to pre-preprocessing of analytic method entities, articles were classified into a set of 15 research disciplines (**Fig 1**) using a supervised machine learning framework pooling information from article titles, abstracts, and journal names. The 15 disciplines were chosen by the authors from a survey of the corpus to balance breadth and specificity of the BLS literature. The disciplines and their abbreviations are as follows: animal/insect/plant biology (ANIMAL), biochemistry and molecular biology (BIOCHEM), clinical research (CLINIC), computer science and informatics (CS), ecology and evolutionary science (ECO), oncology (ONCO), environmental science (ENVIRON),

psychology (PSYCH), population and behavioral genetics (POPGENE), neuroscience (NEURO), chemistry and material science (CHEM), engineering and biotechnology (ENG), human physiology (PHYSIO), immunology (IMMUN), and epidemiology and public health (EPIDEM). The preprocessing pipeline is illustrated in **Fig 1**. Details of the preprocessing pipeline are included in the "Methods and materials" section.

The analytic method entities extracted from all articles were used as input to 3 analytic pipelines: (1) analytic method trends to observe temporal trends in analytic method usage over the time window of 2009 to 2020 (at an annual frequency); (2) discipline by analytic method probability analysis to understand what analytic methods are unique to each BLS discipline; and (3) analysis of analytic method groupings to discover data-driven clusters of analytic methods that frequently co-occur within and across BLS disciplines. To promote reproducibility and reuse, the full code for all preprocessing and analytic processes are provided on the following web page: https://github.com/tsb46/stats_history.

## Journal, discipline, and analytic method statistics

The corpus of open-access peer-reviewed literature predominantly consisted of science general journals, such as *PLOS ONE*, *Scientific Reports*, and *Nature Communications* (**Fig 2A**). This observation highlights one advantage of the machine learning classification of journal articles into scientific disciplines. The common practice of article classification by its journal publication would fail to capture the mixture of scientific disciplines contained within these science general journals. Discipline-specific journals with high article counts included *Oncotarget* (ONCO), *BMJ Open* (CLINIC, EPIDEM), *BMC Genomics* (BIOCHEM), *Sensors* (ENG), *BMC Public Health* (EPIDEM), and *Frontiers in Psychology* (PSYCH). These discipline-specific journals publish peer-reviewed articles in a specific area of study and have a more focused readership. The disciplines with the highest article counts are primarily biomedical and clinical disciplines: CLINIC ($N = 333{,}547$), EPIDEM ($N = 172{,}949$), BIOCHEM ($N = 160{,}016$), and ONCO ($N = 111{,}818$) (**Fig 2B**). The top 10 journals by discipline and the article counts for each discipline are provided in the Supporting information (**S1 Data**).

To provide a visual illustration of the similarity between disciplines in their overall analytic method counts, we deployed classical multidimensional scaling (MDS). MDS is a simple manifold learning technique that expresses each discipline's total analytic method counts in a parsimonious two-dimensional space (**Fig 2B**). The distances between the disciplines in the resulting plot reflect the dissimilarity/similarity in total analytic method counts. This approach made apparent that 2 disciplines stood out as relative outliers in analytic method mentions: Evolution/Ecology and Chemistry/Material Sciences. As depicted in **Fig 5**, these select disciplines revealed a unique profile of analytic method mentions. For illustration, we consider the discipline of Ecology/Evolutionary Sciences. Compared with other BLS disciplines, distance matrix and manifold learning methods (e.g., MDS) are more widely used in the analysis of ecological data [16–18]. Such methods have been found to be uniquely suited for the analysis of species composition and abundance data [19]. For example, distance matrices constructed through metric/nonmetric dissimilarity metrics (e.g., Bray–Curtis dissimilarity) are used to represent a species-by-sample/site matrix. Manifold learning methods are routinely used to analyze the resulting distance matrices [19]. Manifold learning methods are often referred to as "ordination" in ecology.

The most frequently mentioned analytic method entities included null hypothesis testing (e.g., *p*-values and null hypothesis), confidence intervals, correlation, linear regression, logistic regression, *t* tests, and ANOVAs. Other frequently mentioned analytic method entities included bootstrap resampling techniques, meta-analysis, dimension reduction, and clustering

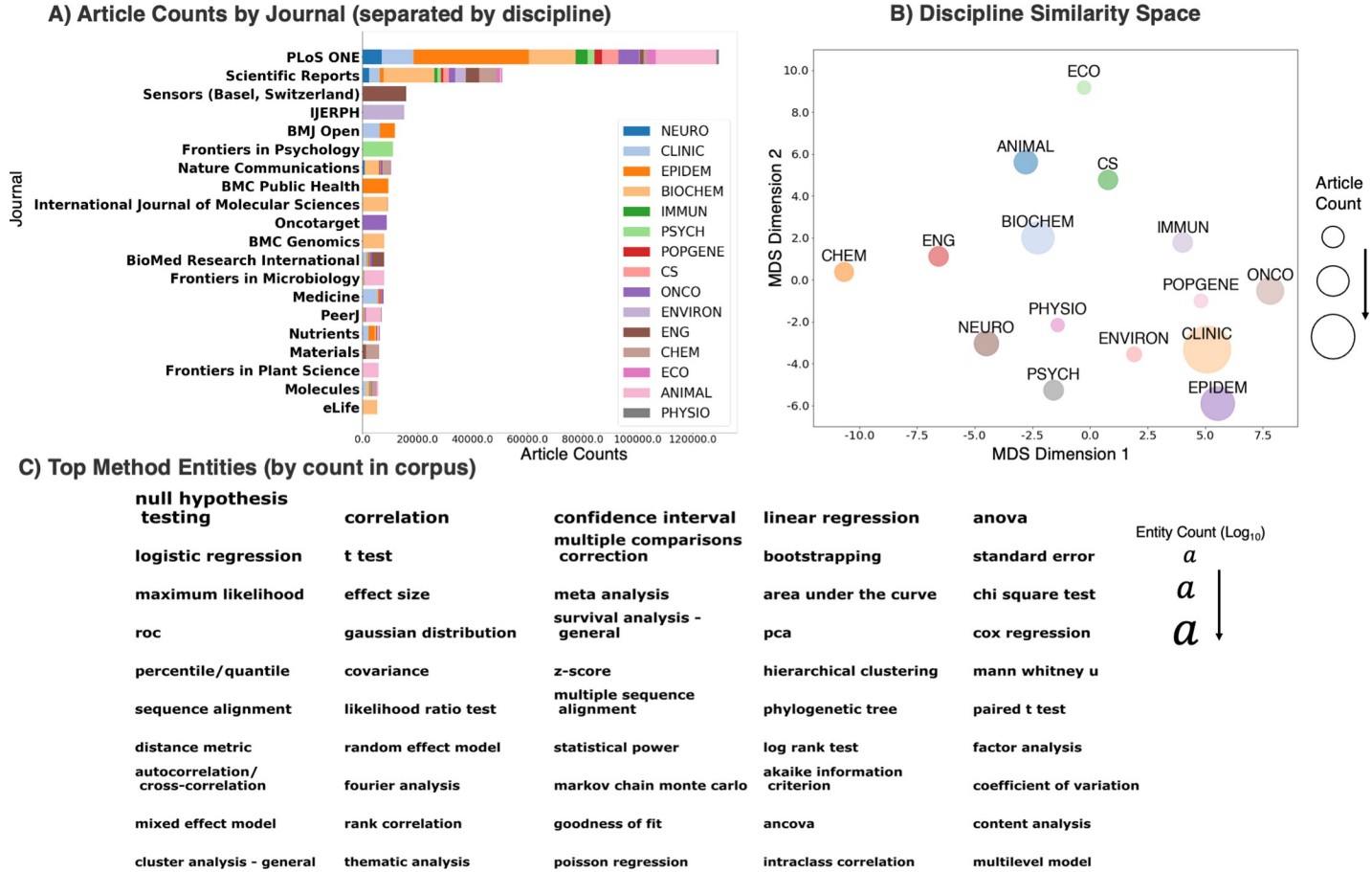

**Fig 2. Journal, discipline and analytic method statistics. (A)** A horizontal stacked bar plot displaying the number of articles for the top 20 journals in the corpus (defined in terms of article count). The percentage of articles per domain within a journal are proportionally shaded within each bar (IJERPH). The research disciplines with the highest article counts were primarily biomedical and clinical disciplines. **(B)** MDS plot displaying the similarity between research disciplines, in terms of total analytic method counts (summed across all articles in the discipline), on a two-dimensional space. The x- and y-axis correspond to the 2 latent dimensions estimated from the MDS solution. The distance between 2 disciplines in this two-dimensional space communicates the dissimilarity in total analytic method counts between the 2 disciplines. **(C)** Top 50 analytic method entities were ranked row-wise by the number of mentions across the corpus. The size of each entity string is proportional to the logged ($log_{10}$) article count. The most frequently mentioned analytic methods were null hypothesis testing, correlation, confidence intervals, and linear regression. Data for all figures are provided in **S1 Data**. ANCOVA, analysis of covariance; ANIMAL, Animal/Insect/Plant Sciences; ANOVA, analysis of variance; BIOCHEM, Biochemistry/Cellular Biology/Molecular Genetics; CHEM, Chemistry/Material Science; CLINIC, Clinical/Hospital Research; CS, Computer Science/Informatics; ECO, Evolution/Ecology; ENG, Engineering/Biotechnology; ENVIRON, Environmental/Earth Science; EPIDEM, Public Health/Epidemiology; IJERPH, *International Journal of Environmental Research and Public Health*; IMMUN, Immunology; MDS, multidimensional scaling; NEURO, Neuroscience; ONCO, Oncology; PCA, principal component analysis; PHYSIO, Human Physiology/Surgery; POPGENE, Population Genetics; PSYCH, Psychology; ROC, receiver operating characteristic.

techniques (e.g., principal component analysis [PCA] and hierarchical clustering), classification performance metrics (e.g., area under the curve and receiver operating characteristic [ROC]), survival models (e.g., cox regression), and bioinformatic algorithms (e.g., sequence alignment and phylogenetic tree construction methods). Interestingly, content analysis—a sometimes quantitative, sometimes qualitative coding method of documents to examine communication patterns—also appears in the top 50 most frequently mentioned method entities.

## Analytic method trends

A primary goal of our survey was to examine trends in analytic method mentions in research articles over the past decade (2009 to 2020). We first manually categorized analytic methods into larger superordinate categories of conceptually similar methods (analytic method

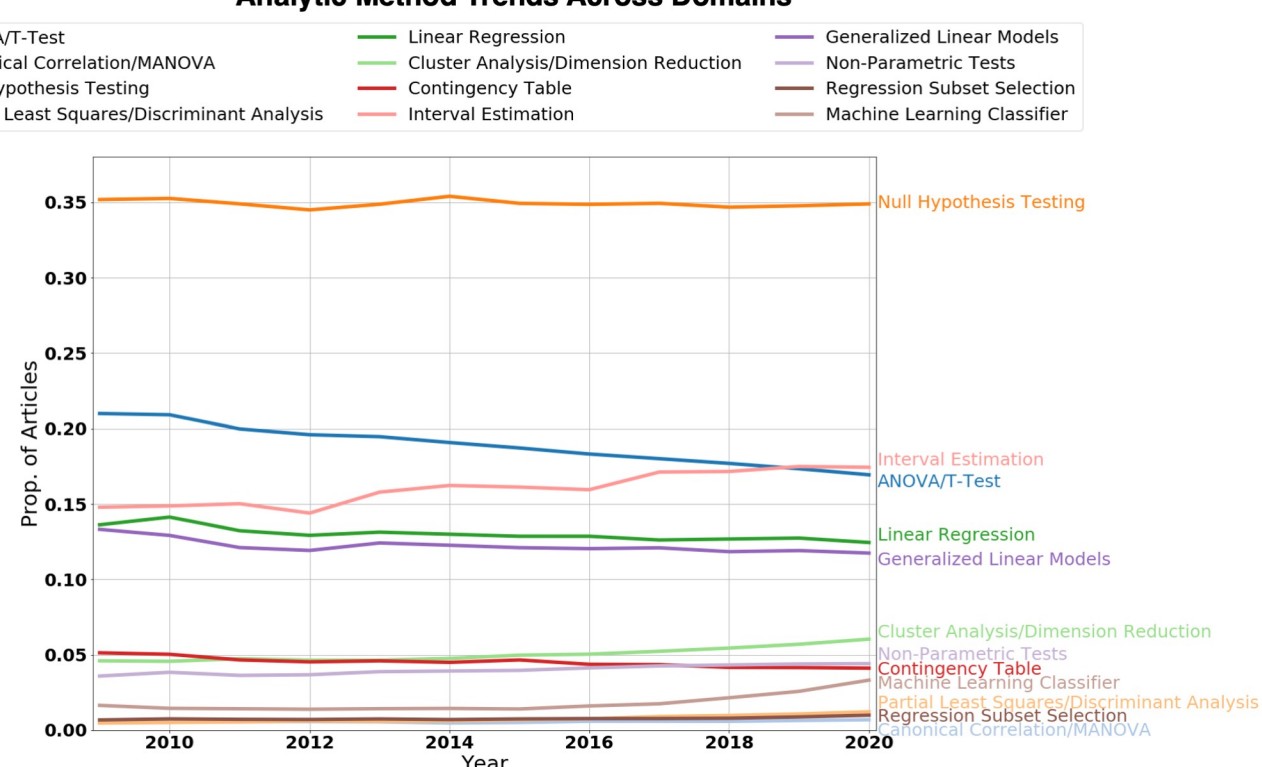

**Fig 3. Overall trends in analytic methods over the past decade.** Time series of 12 analytic method categories from 2009 to 2020 (annual frequency) displayed in a single line plot. For each analytic method category, the time series represents the proportion of articles that contained a mention of that category in their "Methods/ Materials" or "Results" section per year. As can be observed from the plot, baseline differences in the proportion of mentions across analytic method categories are very prominent. Overall, analytic methods taught in introductory research methods and statistics courses (e.g., null hypothesis testing, *t* test/ANOVA, 2-way contingency tables, linear regression, and interval estimation) have been and still are the most mentioned category of analytic methods at the end of the decade (2020). Analysis of individual trends of each analytic method category (**Fig 4**) reveals that while dominant, "introductory" analytic methods have exhibited a decline in mentions over the past decade, while "advanced" analytic methods have shown a consistent increase in mentions. Proportion of article counts by year for all analysis methods are provided in **S2 Data**. ANOVA, analysis of variance; MANOVA, multivariate analysis of variance.

categories—e.g., *t* test/ANOVA, generalized linear models [GLIMs], and survival analysis) (*N* = 34). Because the total number of articles in the corpus increased significantly year over year, the raw frequency counts for all analytic method categories exhibited a consistent increase in total counts over the time span of the corpus. In order to track what analytic method categories have increased or decreased in mentions relative to the total number of articles per year, we calculated the proportion of articles mentioning each category per year (2009 to 2020). We display yearly trends for 12 of the 34 analytic method categories in **Fig 3**. Trends for all analytic method categories are provided in **S1 and S2 Figs**. A data-driven analysis of all analytic method trends without grouping into superordinate categories is provided in **S3 Fig**. Raw counts and proportions for all analytic methods without grouping into superordinate categories are provided in **S1 and S2 Data**, respectively.

We make a distinction between those analytic method categories that are commonly taught in introductory research method and statistics courses—e.g., null hypothesis testing, *t* tests/ ANOVA, linear regression, 2-way contingency tables (e.g., chi-squared test), and interval estimation (e.g., confidence intervals) [20–22] and those analytic method categories taught in advanced statistics or computer science courses—e.g., dimension reduction and clustering techniques (e.g., PCA, nonnegative matrix factorization (NMF), and K-means clustering),

machine learning classifiers (e.g., support vector machines and random forest classifier), and regression subset selection methods (e.g., Lasso regression). We nominally refer to these 2 groups as "introductory" and "advanced" analytic methods, respectively. Note that we do not intend by this distinction to mean that "introductory" methods are less sophisticated or appropriate for data analysis than "advanced" analytic methods. This distinction is merely meant to distinguish between those analytic methods typically taught in introductory courses in data analysis from more advanced undergraduate and graduate data analysis courses.

As can be observed from **Fig 3**, introductory statistical methods—e.g., null hypothesis, testing, *t* tests/ANOVA, linear regression, and confidence intervals—have remained the dominant analytic methods mentioned in BLS methodology and results sections over the past decade. Null hypothesis testing concepts and statistics (e.g., *p*-values, null hypothesis, and alternative hypothesis) are by far the most commonly mentioned analytic "methods" in the BLS sciences —approximately 35% of articles per year mention 1 or more analytic methods in this category. The next most prominently mentioned analytic method category are ANOVA/*t* tests—statistics for comparing mean differences between 1 or more means collected from independent groups or repeated observations. Interval estimation approaches (e.g., confidence interval and credible interval) for calculating the possible values of a population parameter is the third most prominent analytic method category.

While baseline differences in the proportion of articles that mention an analytic method category are important to consider, of primary interest for this survey are the relative trends in mentions across the time span of the study (2009 to 2020). To assess the statistical significance of the linear trends (i.e., decline or increase) in analytic method categories across the study time span, we used a logistic regression model. Specifically, for each analytic method, we modeled the log odds of an article mentioning that method conditional on an article's publication year (2009 to 2020) and the article's scientific discipline. To assess whether any research discipline exhibits a statistically significant deviation from the overall linear trend of a given analytic method category, interactions between the linear trend and discipline were added to the model. To account for the correlated/nonindependent structure of articles published within journals, the logistic regression model was estimated using a generalized estimating equation (GEE) approach. Full details of the model are provided in the Methods and materials section.

The trends in analytic method category mentions are plotted individually (blue) in **Fig 4**. Note that y-axis scales are relative for each analytic method category, and caution should be taken in comparing trends across categories. The statistical significance (*p*-value) of the linear trend for each analytic method category is displayed by each title. Those disciplines that exhibit a statistically significant interaction ($p < 0.05$; corrected for multiple comparisons with the Holm–Bonferroni method)—i.e., exhibit a deviation from the overall trend—are displayed with the overall trend.

While still dominant, there has been a marked decline in the proportion of articles mentioning parametric mean comparison tests (*t* test/ANOVA) over the course of 2009 to 2020. The same decline is observed for analysis of 2-way contingency or cross-tabulation tables, such as the chi-squared test and Fisher exact test. Other "introductory" statistical methods and concepts, such as linear regression and null hypothesis, have remained fairly constant in their proportion of mentions across the past decade. Interestingly, interval estimation methods, such as confidence intervals, have exhibited an increase in mentions over the past decade, perhaps reflecting the increased pressure from institutions and researchers [22–24] to report confidence intervals along with statistical significance tests and *p*-values in peer-reviewed research. As opposed to *p*-values, confidence intervals have the advantage of providing information regarding both the size and uncertainty of a point estimate.

In contrast, analytic methods covered in more advanced statistics and computer science courses have exhibited a marked increase in the proportion of mentions from 2009 to 2020.

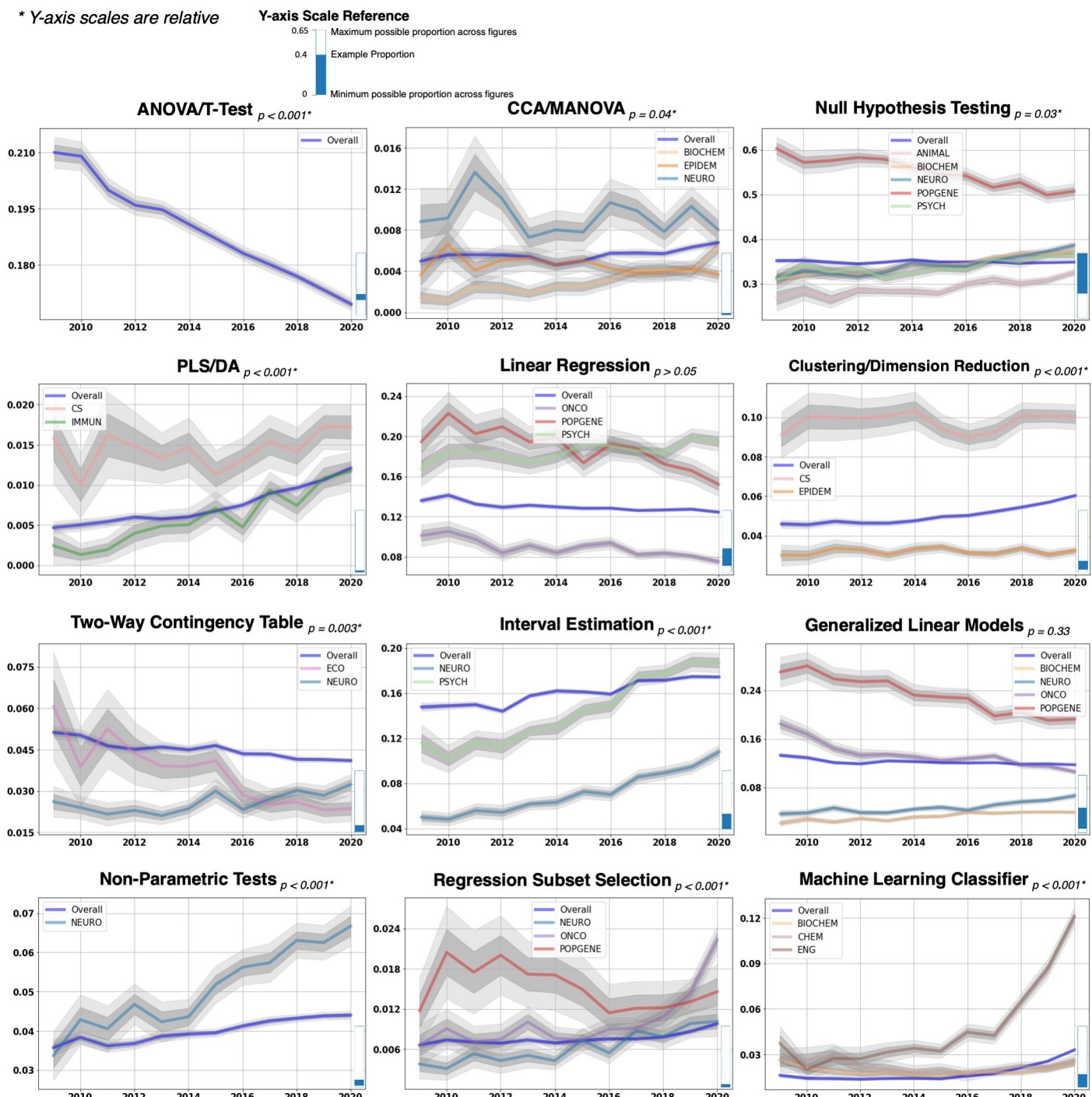

**Fig 4. A decade of analytic method trends (2009–2020).** Time series of 12 analytic method categories from 2009 to 2020 (annual frequency). For each analytic method category, the time series represents the proportion of articles that contained a mention of that category in their "Methods/Materials" or "Results" section per year. The time series of each analytic method category is displayed in its own plot with different y-axis scales. Note that because each plot differs in y-axis scale, caution should be observed when comparing trends across categories. To help readers compare y-axis scales across categories, we have provided a y-axis scale bar in the bottom right of each figure. An illustration of the y-axis scale bar is presented at the top of the figure. The height of the bar corresponds to the distance between the minimum possible proportion (0) and the maximum possible proportion (0.65) of all articles per year across all categories. The highlighted region of the bar (in blue) corresponds to that categories' range of proportion values (across years) from 0 to 0.65. Random sampling variability for each proportion estimate was visualized using bootstrapped SEs from 100 bootstrapped samples of articles at each time point (dark shaded region: ± 1 SE, light shaded region: ± 2 SE). Overall, analytic methods taught in introductory research methods and statistics courses (e.g., *t* test/ANOVA, 2-way contingency tables, and linear regression) have shown differing

rates of decline or remained stable in mentions over the past decade, with the exception of interval estimation approaches (e.g., confidence intervals). On the other hand, advanced analytic methods (e.g., machine learning classifier, regression subset selection methods, and clustering/dimension reduction) have shown a consistent increase in mentions over the past decade. Proportion of article counts by year for all analysis methods are provided in **S2 Data**. Python code for modeling trends of analysis methods is provided at https://github.com/tsb46/stats_history/blob/master/demo.ipynb. ANOVA, analysis of variance; CCA, canonical correlation analysis; MANOVA, multivariate analysis of variance; PLS/DA, partial least squares/discriminant analysis; SE, standard error.

These analytic methods include dimension reduction/clustering analysis, machine learning classifiers (e.g., random forest classifiers, support vector machines, and artificial neural networks [ANNs]), nonparametric tests, partial least squares/discriminant analysis (PLS/DA), and regression subset selection methods (e.g., Lasso regression). Generalized linear models (GLIMs) (e.g., logistic regression, probit regression, and Poisson regression) have remained relatively constant over the past decade. However, analysis of the individual GLIM models belonging to this category shows that logistic regression, perhaps the most common GLIM model, has declined in mentions over the past decade (**S2 Data**).

As illustrated in **Fig 4**, not all research disciplines follow the overall trend in analytic method mentions over the past decade. Some research disciplines exhibit a trend opposite of that observed in the overall trend. For example, mentions of null hypothesis testing concepts in population/behavioral genetics (POPGENE) research articles have declined in mentions from 2009 to 2020, while the overall trend remains fairly constant during that time span. Other research disciplines exhibit a steeper trend in mentions than that observed in the overall trend. For example, mentions of machine learning classifier methods in engineering/biotechnology (ENG) research articles exhibit a much steeper exponential increase in mentions from 2009 to 2020 compared with the overall trend.

## Analytic method mentions by discipline

To examine analytic methods associated with each BLS discipline, we used a contingency table approach. Specifically, we modeled the difference in observed versus expected article counts based on the number of article counts for each discipline and analytic method across the corpus. We used the standardized Pearson chi-squared residuals as an effect size measure of the degree to which an analytic method is more prominently associated with a given discipline relative to other disciplines. In **Fig 5**, we display the top 10 analytic methods per discipline as measured by the chi-squared residual value. The raw probability for each analytic method by discipline is provided in the Supporting information (**S3 Data**).

As can be observed from **Fig 5**, each discipline is associated with a distinct set of analytic methods. Some analytic methods appear across more than 1 discipline—e.g., Fourier analysis in chemistry/material sciences and engineering/biotechnology. Others are unique to a given discipline. For example, mentions of independent component analysis (ICA) appear much more in neuroscience research articles than those of other disciplines. ICA is a common method for decomposing multivariate signals (typically time series) into an additive mixture of statistically independent latent sources, which has found common use in neuroimaging (e.g., functional magnetic resonance imaging and electroencephalography) for source separation, artifact removal and detection of brain networks [25,26]. Another example is survival analysis methods (e.g., Cox regression and log-rank test) in oncology. Survival analysis methods, such as Cox regression or Kaplan–Meier analysis, aim to model the time to an event of interest. For clinical trials in the discipline of oncology, survival analysis methods have been found useful in modeling the effect of treatment on time to death [27]. Another example is the uniquely predominant use of PLS and PLS/DA in chemistry or chemometrics. PLS, a multivariate technique that predicts a set of response variables (Y) based on a set of predictor variables

**NEURO** 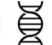

ica
multiple comparisons correction
anova
t test
signal filtering
paired t test
graph theory - network statistic
hilbert transform
granger causality
z-score

**CLINIC** 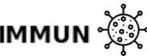

confidence interval
logistic regression
meta analysis
null hypothesis testing
roc
area under the curve
random effect model
chi square test
multiple imputation
cox regression

**EPIDEM** 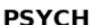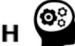

logistic regression
confidence interval
thematic analysis
content analysis
poisson regression
percentile/quantile
multinomial logistic regression
grounded theory
linear regression
chi square test

**BIOCHEM** 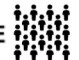

sequence alignment
hierarchical clustering
multiple sequence alignment
multiple comparisons correction
t test
hypergeometric test
anova
phylogenetic tree
null hypothesis testing
empirical bayes

**IMMUN** 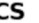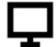

multiple sequence alignment
phylogenetic tree
bootstrapping
sequence alignment
geometric mean
maximum likelihood
markov chain monte carlo
gamma distribution
mann whitney u
fishers exact test

**PSYCH** 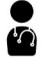

structural equation modeling
effect size
confirmatory factor analysis
multilevel model
factor analysis
kurtosis
path analysis
maximum likelihood
skewness
lineweaver burk plot

**POPGENE** 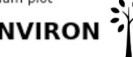

manhattan plot
meta analysis
multiple comparisons correction
statistical power
fixed effect model
logistic regression
type i error
effect size
qq plot
linkage disequilibrium plot

**CS** 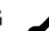

neural network
classification accuracy
support vector machine
random forest decision tree
gradient descent
mutual information
decision tree
graph theory
bayesian posterior
graphical model

**ONCO** 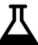

cox regression
survival analysis - general
log rank test
confidence interval
kaplan meier analysis
roc
null hypothesis testing
fixed effect model
area under the curve
random effect model

**ENVIRON** 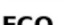

geometric mean
linear regression
percentile/quantile
regression spline
gaussian process regression
arma/arima
structural equation modeling
confirmatory factor analysis
poisson regression
autocorrelation/cross-correlation

**ENG**

fourier analysis
kalman filter
neural network
signal filtering
correlation
classification accuracy
wavelet analysis
support vector machine
uniform distribution
autocorrelation/cross-correlation

**CHEM**

fourier analysis
correlation
uniform distribution
r squared
pca
simulated annealing
partial least squares - discriminant analysis
genetic algorithm
autocorrelation/cross-correlation
partial least squares

**ECO** 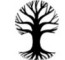

markov chain monte carlo
bootstrapping
maximum likelihood
mantel test
multiple sequence alignment
akaike information criterion
phylogenetic tree
mds
gamma distribution
distance metric

**ANIMAL** 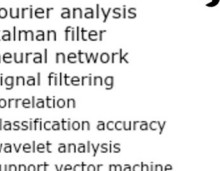

phylogenetic tree
sequence alignment
multiple sequence alignment
bootstrapping
mds
anova
maximum likelihood
hierarchical clustering
redundancy analysis
analysis of similarity

**PHYSIO** 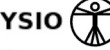

intraclass correlation
effect size
anova
shapiro wilk test
paired t test
coefficient of variation
repeated measures anova
detrended fluctuation analysis
bland altman analysis
ancova

**Fig 5. Analytic method mentions across disciplines.** Top 10 standardized chi-squared residuals for each discipline from the contingency table analysis, ranked from top to bottom. The font size of the analytic method string is sized by its (logged) standardized chi-squared residual. The greater the chi-squared residual, the greater the difference between the observed and expected number of analytic method entities within that discipline. Disciplines have a unique set of analytic methods frequently mentioned in their subject matter, e.g., neural networks in computer science, ICA in neuroscience, and Manhattan plots in population/behavioral genetics. Standardized chi-squared residuals for all analysis methods by domain are provided in **S3 Data**. ANOVA, analysis of variance; ARMA, autoregressive moving average model; ICA, independent component analysis; MDS, multidimensional scaling; ROC, receiver operating characteristic; PCA, principal component analysis.

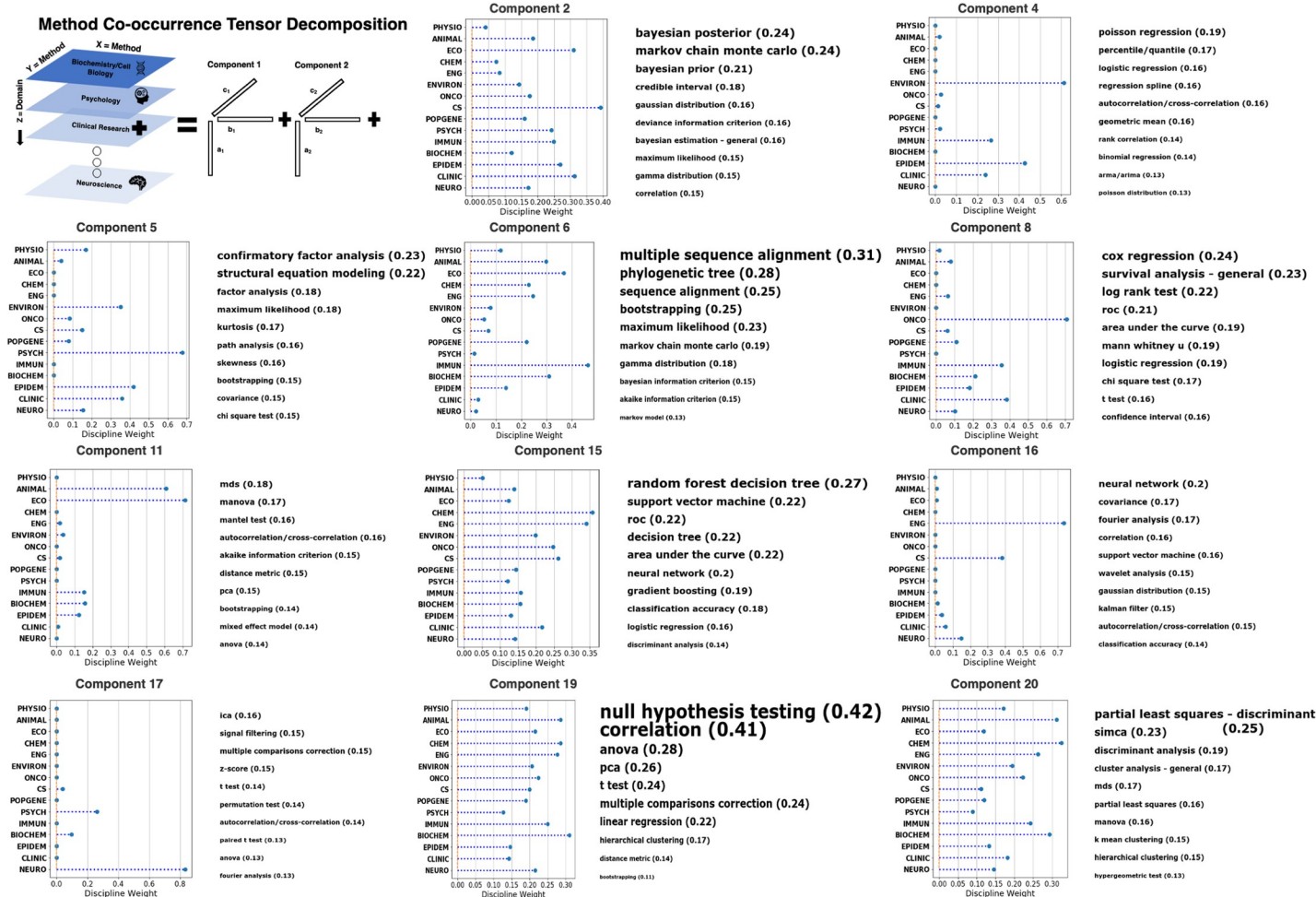

**Fig 6. Co-occurrence patterns in article mentions of analytic method across BLS disciplines.** To understand what analytic methods are frequently used together in the same study, we conducted a tensor decomposition of an analytic method co-occurrence by discipline tensor. The tensor decomposition analysis simultaneously models the co-occurrence between analytic methods, as well as their frequency of mentions in each discipline. This figure displays the discipline and analytic method weights from the tensor decomposition analysis. Components from the tensor decomposition are referred to as "method groupings" or groups of analytic methods that frequently occur together in study method and results sections. The top left panel provides a visual illustration of the tensor decomposition (nonnegative CANDECOMP decomposition) of the analytic method co-occurrence by discipline tensor. The first 2 dimensions of the tensor represent the logged sum of co-occurrences between each pair of analytic methods. The third dimension splits out the analytic method co-occurrences by discipline (i.e., the analytic method co-occurrences of articles within each discipline). For each component or "method grouping," a stem plot illustrates the weights for each discipline, as well as the top 10 analytic methods, in terms of their weights (sized by their weight). For each component, the discipline weights represent the frequency of usage of that component across each discipline. Some sets of analytic methods are represented across all BLS disciplines (e.g., component 19), while others are concentrated within 1 or 2 disciplines (e.g., component 17). Discipline and analysis method weights for all 20 components are provided in **S4 Data**. ANOVA, analysis of variance; BLS, biomedical, life, and social; MANOVA, multivariate analysis of variance; MDS, multidimensional scaling; PCA, principal component analysis; ROC, receiver operating characteristic.

(X), is often used in chemometrics to relate properties of chemical samples (e.g., spectral properties) to their chemical composition (e.g., sample concentrations) [28].

## Analytic method groupings

Analytic methods in the BLS sciences are rarely used in isolation. Rather, a set of methods are applied jointly, or in sequence, to understand a dataset. We term these frequently co-occurring analytic methods, "method groupings." To directly extract coherent constellations of methods and understand how they vary across BLS disciplines, we applied a tensor decomposition approach to an analytic method ($N = 250$) co-occurrence by discipline ($N = 15$) tensor (**Fig 6**;

top left panel). For illustration, we displayed 11 components (method groupings) of a 20 component (i.e., 20 rank-one tensors) solution (**Fig 6**). Visualization of all component weights is provided in **S4 Fig**. Each component is associated with separate weights for analytic methods and disciplines, indicating the analytic methods and disciplines most associated with the component, respectively.

The method groupings revealed by the tensor decomposition can be roughly classified into cross-discipline and within-discipline method groupings (i.e., components). Cross-discipline method groupings include components with a broad representation across the BLS disciplines (relatively more even distribution of discipline weights). For example, components 2, 6, 15, 19, and 20 had nonzero weights for the majority of BLS disciplines. Component 2 included methods broadly related to Bayesian statistics and concepts, such as credible intervals and posterior/prior probability distributions (e.g., beta distribution). This component is prominent across all research disciplines with particular prominence in computer science, ecology, and clinical research disciplines. Component 6 included bioinformatic algorithms for DNA sequence alignment, phylogenetic tree construction, and bootstrap resampling. This component is more prominent in biological science disciplines, such as ecology and evolution, animal and plant sciences, immunology, and biochemistry and molecular biology. Component 15 includes a variety of machine learning algorithms/metrics, including random forest algorithms, support vector machines, ANNs, and classification accuracy metrics (e.g., ROC and area under the curve). This component is prominent in all research disciplines with a particular prominence in chemistry and material sciences, and computer science.

We subsequently focused attention on the discipline-specific method groupings: method groupings with almost exclusive use in 1 or a small subset of BLS disciplines. One example of a discipline-specific method grouping was component 11, a component exclusively represented in ecology and evolution and animal and plant disciplines. This set of methods included MDS, ANOVA-based methods, and distance matrix analyses (e.g., Mantel test). As noted above, manifold learning and distance matrix methods are uniquely suited to analyses of species composition and other types of data regularly collected in these disciplines. The appearance of ANOVA-based methods in this seemingly unrelated group of methods may seem surprising, but owes to the fact that variance partitioning of distance matrices is a historically common practice in ecological disciplines [29]. Another discipline-specific method grouping is component 16 represented most prominently by computer science and engineering and biotechnology disciplines. The analytic method with the strongest weight for this component is ANNs. So-called "deep learning," ANNs with many layers of nodes, have seen an explosion of interest in recent years due to the increase in computational power and big data sources. Consistent with these findings, computer science, bioinformatics, and engineering/biotechnology disciplines have been at the forefront of methodological development in this area [30]. Another discipline-specific method grouping is component 5, consisting of latent variable models, including structural equation modeling (SEM) and confirmatory factor analysis (a subset of SEM). This component is most prominent in the discipline of psychology. Latent variable methods have found particular use in the field of psychometrics, where these models have been used to measure theoretical constructs, such as intelligence, personality, and attitudes [31].

## Discussion

The data analytic landscape of the BLS sciences is subject to change. The democratization and commoditization of tools for quantitative analysis have grown exponentially in the 21th century and only accelerated in pace in the past decade. This tectonic shift is due to the increased

accessibility of computational resources, open-source software and abundance of big data in more areas of human activity. When learning to conduct data analysis, the scientist in training is faced with a steep hill to climb. To make this climb easier, graduate and undergraduate education must reflect the current practices and trends in data analysis. We offer an automated 12-year survey of approximately 1.3 million open research papers to characterize the data analytic landscape of the BLS sciences. This study aimed to provide a snapshot of the ongoing methodological shifts across a variety of scientific communities.

We find that the analytic methods commonly taught in introductory research methods and statistics courses (e.g., *t* test and ANOVA) remain the most commonly mentioned methods in "Methods" and "Results" sections of research articles. However, while dominant, these methods have largely declined in prominence, or remained stable, over the time span of the study (2009 to 2020). On the other hand, multivariate statistics and machine learning methods have exhibited a consistent, sometimes exponential, increase in mentions over the time span of the study. Further, we find that certain analytic methods are not equally distributed across BLS disciplines, but have unique prominence in certain disciplines over others. We believe our results provide valuable insights into how university curricula should be designed to meet the urgent need for training a new generation of quantitatively literate scientists.

"Multivariate statistics and machine learning methods" is a broad label, referring to a wide variety of analytic methods. These methods are often taught in advanced statistics and computer science courses and include PCA, regression subset selection, PLS, support vector machines, random forest algorithms, and ANNs. While some of these methods are quite old (e.g., PCA was first developed in the early 20th century), others are relatively new and still developing (e.g., ANNs have only seen broad use in the past decade). This study made no attempt to examine the potential causal factors behind the observed trends. We speculate that they could be due to several reasons: (1) the collection of larger and more complex datasets; (2) the recent popularity of data science as a tool in academia and industry; or (3) an increasing realization among researchers that manuscripts containing advanced analytics are more likely to impress reviewers and editors. However, advanced analytic methods still only represent a minority of the mentions we observed across BLS research articles. Although on different rates of decline or stability, the statistical methods taught in introductory courses, such as mean comparison tests (e.g., *t* test/ANOVA), cross-tabulation/contingency table analysis (e.g., chi-squared test), and null hypothesis testing, represent a much more sizable percentage of mentions across BLS research articles.

Projecting these trends into the future, we would expect that multivariate statistics and machine learning methods will enjoy increasing usage relative to more traditional statistical testing frameworks. The traditional statistical tool stack—*t* test, ANOVA, z-test, etc.—taught in undergraduate and graduate statistics courses were largely developed in the first half of the 20th century [32]. Since then, advances in computation and statistical computing software have revolutionized the analytic tools available to researchers. The application of advanced multivariate and machine learning analysis methods requires proficiency in statistical software and/or open-source programming languages that are largely absent from most traditional research methods and statistics curricula. Overall, these trends suggest that introductory research methods and statistics courses may benefit from incorporating a "data science" focus in their curricula [8,32].

Our survey demonstrates that methods for data analysis can vary widely across BLS disciplines. Several explanations can be offered for the distinct usage of data analysis methods between BLS disciplines. Perhaps the primary driver of a discipline's adoption of research methods is the simple observation that the subject matter lends itself to the assumptions and goals of selected analytic methods. For example, consider the observed disproportionate use of

SEM in the discipline of psychology (**Figs 5 and 6** –component 5). Psychological research routinely relates observable behavior such as task performance or questionnaire responses to unobserved or latent variables. The desire to explore causal structure among these latent variables has led to the systematic adoption of SEM—a technique to specify and test causal structures among latent and observable variables [33]. Similar explanations can be offered for other method–discipline pairs, such as survival models and oncology. Other differences may arise from historical contingency, with no necessary connection between an analysis method and the subject matter it is applied to. For example, consider the predominant use of Fisher exact test in immunology versus the chi-squared test in clinical research (**Fig 5**). Both are statistical significance tests of the association between 2 categorical variables. The appropriate context for each test is controversial among statisticians, but Fisher exact test is commonly recommended over the chi-squared test with small sample sizes [34,35]. Despite this controversy, our analysis indicates the Fisher exact test is generally preferred over the chi-squared test in the field of immunology and vice versa in clinical research. Thus, one might hypothesize that either (1) the discipline of immunology works with smaller sample sizes on average than the discipline of clinical research; or (2) these differences arose for sociological or historical factors.

Differences in analytic method usage have concrete implications for the direction of research in each BLS discipline. The choice of experimental or observational design often entails the subsequent analytic method used to analyze the data, but a reverse influence occurs as well: The researcher's knowledge of available analytic methods informs their experimental or observational design. For example, ANOVA models for analysis of group means have a historically close relationship with experimental design in social and life science research [36]. In other words, the influence between the choice of data analysis method and how data is collected operates in both directions. This observation underlies the potential for cross-fertilization and mutual inspiration between BLS disciplines by the discovery of new methods for data analysis, as well as novel ideas around data collection. While many advocates of cross-disciplinary collaboration have emphasized the joining together of different theoretical and subject matter expertise [37], our findings emphasize a further methodological benefit of collaboration, which affords practitioners access to novel methods of data analysis not yet widely known in their own disciplines.

It should be noted that the corpus and methods used in this analysis are limited in many respects. First and foremost, our study relies on mentions of an analytic method in a research article as an indicator for the usage of that analytic method in the article. However, the mention of an analytic method is not an unequivocal indication of its usage for data analysis in a research article. Therefore, we restricted our analysis of research articles to their "Methods" and "Results" sections, where mentions of an analytic method should be more closely associated with their usage. However, due to the infeasibility of spot-checking a large corpus (approximately 1.3 million articles) manually, there may be cases of analytic method mentions that did not imply their usage in our sample. Further, the entity recognition approach in our study requires that analytic methods are explicitly reported in methodology and results sections. However, inadequate and/or inaccurate reporting of protocols and statistical analyses is a known problem in the BLS sciences [38–40]. Thus, there are likely a number of research articles in our corpus where the full set of analytic methods employed was not captured by our analysis. Second, the usage of analytic methods does not imply that the method was applied appropriately or correctly. In fact, the inappropriate application of statistical methods may be a contributor to the replication crisis in the BLS sciences [23,41,42]. Thus, the increase in usage of multivariate and machine learning methods should not be considered prima facie evidence that these methods are being used appropriately. Third, our corpus only contains open-

access articles made available by an open-access journal or an NIH-funded author. Thus, a sizable collection of peer-reviewed research in the past decade is systematically missing from this analysis. However, we assume that the type of publisher—open access or subscription based—is not a significant determiner of the methods used within a discipline. Fourth, some scientific disciplines may be less well represented in this survey, including experimental and theoretical physics, anthropology, astronomy, cosmology, economics, sociology, and geology. Future studies with a more comprehensive corpus of scientific publications will provide deeper insight into the historical and cross-disciplinary trends in scientific data analysis. Fifth, our classification of scientific disciplines requires that an article is assigned to one, and only one discipline. While the majority of research articles in our corpus may fit into one scientific discipline over others, some will be multidisciplinary, particularly for those disciplines with similar research agendas and regular collaboration (e.g., oncology, clinical research, and immunology). We have provided the original Python code of all preprocessing and analytic pipelines for those who wish to improve or redesign this study's algorithms for future use (https://github.com/tsb46/stats_history).

Our data-driven survey of peer-reviewed articles reveals that the analytic landscape of the BLS sciences has been transformed over the past decade. A comparable rate of change will be required in education of budding scientists in statistics and research methodology. Equally important is the observed analytic diversity of BLS sciences of the past 10 years. The diverse analytic tool sets across BLS disciplines promises large payoffs for cross-disciplinary collaboration. In this vein, the recent advent of big data and open-source science is at least as much an opportunity for adequately training the next generation of researchers, as a challenge.

## Methods and materials

### Peer-reviewed literature corpus

Two sources of peer-reviewed literature were used for this analysis: the PMC OAS ($N$ = 3,300,810 articles at time of study) and the PubMed Central Author Manuscript (PMC AM) collection ($N$ = 702,198 articles). The PMC OAS provides access to full texts from a total of 14,722 open access peer-reviewed journals (at time of study). The PMC AM collection provides access to full texts of manuscripts made available in PMC by authors in compliance with the NIH Public Access Policy. Both sources form part of PMC's open access collection [15] (https://www.ncbi.nlm.nih.gov/pmc/tools/textmining/). Bulk downloads of the full OAS and AM collection articles were conducted using the PMC FTP service. Overall, a total of approximately 4 million articles were downloaded and screened for our analysis.

### Article parsing and method section identification

Both OAS and AM text corpora are openly available in a structured and standardized form to the public on PMC's web page. Both corpora were downloaded in XML format. Each XML article file is organized into article metadata and article text separated into article sections (e.g., Introduction, Methods and Materials, Results, etc.). As the primary interest of this study was exploring usage patterns of analytic methods in this pool of articles, we retrieved sections of the XML articles that correspond to the methodology (e.g., "Methods and materials" section) and results section of the article. Given that methodology and results sections have no standardized title, we pulled any sections of text from each XML article that contained in any of the following sequence of strings: "method," "material," "measure," "analysis," "statistical," "model," "algorithm," and "result." As we were interested in original studies, XML articles not containing any of these section search strings were excluded from analysis, such as literature

reviews, book reviews, commentaries, etc. Of the total XML articles in the peer-reviewed literature corpus, $N_{text}$ = 2,292,668 articles with the above section search strings were retrieved.

## Named entity recognition of analytic method entities

In the resulting corpus of methodology and results section texts, it would be infeasible to manually tag each analytic method phrase in every section text. Thus, we opted for an automated recognition method for the purpose of detecting analytic method phrases. We used a combined phrase matching/rule-based and machine learning approach. First, we developed a large list of phrases ($N_{phrase}$ = 1,129) corresponding to commonly used analytic methods across BLS disciplines. These were used as a rule-based matching approach to detect analytic method phrases in the section texts. To generalize detection of analytic method phrases outside our manually created list, we selected 20,000 random method and results section texts with tagged phrases from our rule-based approach and fed them as training examples to a statistical NER algorithm. The objective of the NER algorithm is to utilize the rule-based training samples as context "clues" for detecting analytic method phrases more generally (i.e., those outside the original phrase list). To perform NER on our full corpus, we used the convolutional neural network (CNN) algorithm provided by the open-source spaCy python package [38], with standard parameters for training (100 iterations, 0.2 dropout, and mini-batch training). The trained NER model was applied to the entire corpus of methodology and results section texts to generate the final list of detected analytic method phrases or entities. Of the total corpus of 2,292,668 methodology section texts, at least 1 analytic method entity was detected in over half of the texts ($N_{text}$ = 1,438,077). In the main text, we refer to the detected analytic method phrases from the trained NER model as analytic method entities, or simply, analytic methods. The total number of unique method entities (before pre-preprocessing) discovered from the NER algorithm was $N_{entity}$ = 15,016.

## Analytic method entity preprocessing

The NER algorithm yielded a unique list of analytic method entities, which was represented as a distinct sequence of characters. Different entities can refer to the same analytic method, for example, one could refer to a *t* test as "*t*-tests," "*t* test," "ttest," etc. To attempt to correct for small spelling differences such as these, each analytic method entity string underwent a sequence of preprocessing steps for harmonization: (1) lowercase characters; (2) removal of non-alphanumeric or non-Greek characters (e.g., hyphenations, quotes, and commas); (3) lemmatization of words of the tokenized (i.e., separated into words) entity strings; and (4) converting entity strings that occur rarely ($N$ < 5) to more commonly used spellings (within a max difference of 2 characters) using the SymSpell algorithm implemented in https://github.com/wolfgarbe/symspell. As another measure of quality control, only preprocessed entities with a minimum of 10 occurrences were included in the final method entity vocabulary. The final quality control step was a manual entity disambiguation procedure. Most analytic methods can be referred to by different names. For example, an independent sample *t* test may be referred to as an "independent *t*-test," "two-sample *t*-test," "Student's two-sample *t*-test," and so forth. Cox proportional hazards regression may be referred to as "Cox regression," "Cox PH regression," "proportional hazards regression," and so forth. To ensure that differences in analytic method counts across the corpus do not reflect different naming conventions, the authors conducted a thorough review of all preprocessed entity strings and categorized those strings into similar categories. In addition, "junk" entity strings that were mistakenly identified as analytic methods by the NER algorithm were thrown out in this stage. The final method

entity vocabulary count after preprocessing was $N_{entity}$ = 250. After preprocessing, the final pool of articles was $N_{text}$ = 1,364,517.

## Article classification into research disciplines

A primary objective of this study is to understand cross-disciplinary usage in analytic methods. This objective requires that the articles in the corpus are first classified into separable disciplines (e.g., biochemistry, epidemiology, psychology, etc.). A simple approach would be to manually classify each journal (or publication) in our corpus into a discipline, such that each article belonging to that journal would be classified with that discipline. While a much more manageable task than manually classifying each article, this approach runs into 2 problems: (1) not all articles in a domain-specific journal (e.g., *PLOS Biology*, *Medicine*, and *EMBO Journal*) can be easily classified into 1 BLS discipline; and (2) domain general journals (e.g., *PLOS ONE*, *Nature Communications*, and *Science*) cannot be classified into a single BLS discipline. Thus, we chose to use a machine learning text classification approach to classify each single article into a set of prespecified BLS disciplines. This approach allowed for more flexible classification at the level of each journal article—allowing for different article classifications within the same journal.

We chose a set of 15 BLS discipline categories for article classification: Biochemistry/Cellular Biology/Genetics (BIOCHEM), Clinical/Hospital Research (CLINIC), Computer Science/Informatics (CS), Chemistry/Material Sciences (CHEM), Environmental/Earth Science (ENVIRON), Evolution/Ecology (ECO), Immunology (IMMUN), Oncology (ONCO), Psychology (PSYCH), Neuroscience (NEURO), Public Health/Epidemiology (EPIDEM), Engineering/Biotechnology (ENG), Human Physiology/Surgery (PHYSIO), and Population Genetics (POPGENE). Importantly, we make no claim that this categorization represents the most optimal division of BLS disciplines—a potentially infinite number of categorizations could be more/less useful in certain contexts and overlap in research topics between the categories of any division will be prevalent. Rather, this categorization provides a useful/pragmatic division of BLS disciplines given the distribution of publications in our corpus. To classify articles in the corpus into the 15 BLS disciplines, we input a bag-of-words feature set (1-gram, 2-gram, and 3-gram tokens), generated from the article abstract, title, and journal title, to a multinomial naive Bayes (MNB) classification algorithm. We utilized a version of the MNB algorithm that corrects for an unequal number of instances across categories [43], as was present in our corpus. To reduce the number of features input to the MNB algorithm and improve prediction accuracy, we applied a chi-squared feature selection approach. Specifically, we chose the top 15,000 features from the bag-of-words feature set, after ranking the full set of features based on their chi-squared statistic value with the 15 discipline categories. The chi-squared statistic is a measure of dependence between 2 categorical variables, and in our case, provides an indication of the degree to which a text token (e.g., 1-gram, 2-gram, or 3-gram) is associated with differences in scientific disciplines. The final MNB model for training involved 15,000 features, 15 discipline categories to be distinguished, and 1,470 training samples. We assessed the model accuracy using a repeated K-fold cross-validation approach ($N_{Folds}$ = 10, $N_{Repeats}$ = 10). The final MNB model achieved an accuracy of 74.9%. The accuracy achieved for each individual discipline are as follows: ANIMAL—71.2%, BIOCHEM—72.2%, CHEM—88.4%, CLINIC—77.6%, CS—79.2%, ECO—75.3%, ENG—67.4%, ENVIRON—67.8%, EPIDEM—65.6%, IMMUN—64.0%, NEURO—87.1%, ONCO—84.5%, PHYSIO—69.6%, POPGENE—60.0%, and PSYCH—69.2%. We found that the MNB approach performed better than other classification approaches (e.g., random forest decision tree classification, support vector machines, and logistic regression) on our dataset, in terms of classification accuracy.

## Discipline similarity analysis

To examine the similarity of analytic method mentions between disciplines, we used an MDS approach. First, we summed the counts of all analytic method entities ($N$ = 250) per discipline. We then computed the Euclidean distance between all pairs of z-score normalized count vectors (1 vector per discipline). The distances were then projected to a two-dimensional space using a classical MDS algorithm.

## Analytic method trend analysis

In order to understand temporal changes in analytic method mentions across BLS sciences, we first manually categorized analytic methods into larger superordinate categories of conceptually similar methods (analytic method categories—e.g., $t$ test/ANOVA, GLIMs, and survival analysis) ($N$ = 34). We then computed annual counts of analytic method categories starting at the beginning of 2009 to the end of 2020 ($N$ = 12 time points). We chose an annual time resolution for the following reasons: (1) some journals in our corpus are published at a limited time frequency (e.g., annually or biannually); and (2) it was reasoned that large-scale trends in analytic method usage within a discipline occur at a low frequency (i.e., over years, as opposed to months). We took a model-based approach to test the statistical significance of the linear trend of each analytic method category. Specifically, we modeled the log odds of an article mentioning an analytic method category across the 12-year time span using a logit model. The statistical significance of a (yearly) linear trend term was assessed at a $p < 0.05$ (corrected for multiple comparisons with the Holm–Bonferroni method). In addition, sum-coded (deviation-coded) variables were included for the discipline the article belonged to ($N = 15 − 1$, predicted from the classifier mentioned above), where clinical research was used as the omitted reference level. To examine whether any discipline's linear trend for a given analytic method category deviated from the overall trend, we assessed the statistical significance of the interaction between each discipline-coded effect variable and the linear trend term. The coefficient of each discipline-by-linear trend interaction represents the degree to which that discipline significantly deviates from the overall linear trend. Our corpus contains a prominent multilevel structure with articles nested within journals, such that we would expect analytic method counts between articles within the same journal to be more similar than across journals. To account for this nested dependence structure in the corpus, we estimated the logit model using Generalized estimating equations (GEE). The GEE estimated model estimates a population-averaged linear trend correcting the standard errors for the nested dependence structure, as opposed to a within-journal effect (i.e., mixed-effects model).

The logit models for each analytic method category were used primarily to estimate the statistical significance of the linear trends and possible discipline–trend interactions. For the main text, time series for each analytic method are presented visually in line plots (**Figs 3 and 4**). The annual article count in our corpus increased at approximately a linear frequency from 2009 to 2020: 2009: 34,666; 2010: 45,915; 2011: 55,611; 2012: 70,008; 2013: 93,461; 2014: 109,828; 2015: 122,138; 2016: 134,136; 2017: 146,029; 2018: 159,920; 2019: 181,419; and 2020: 211,386. To account for this increase in article frequency, we normalized the raw frequency counts of each analytic method category by the total article counts per year—i.e., we computed the proportion of counts of each method category to the total article counts per year. Thus, all analytic method category trends are displayed as proportions of the total number of articles per year. To provide an estimate of uncertainty of the proportion of mentions for a given analytic method category and year, we obtained an estimate of the sampling variation by constructing 100 bootstrapped samples of all articles within a year and recalculated the proportion by total article count for that method category. Standard errors were estimated from the

standard deviation across the 100 bootstrapped proportions and displayed as shaded regions around the observed proportion trends.

## Discipline by analytic method probability analysis

In order to understand the unique clustering of analytic method entities between disciplines, we used a chi-squared contingency table approach. This approach provides a straightforward way to identify those analytic methods that are used more frequently in a discipline relative to others. Specifically, we organized the data into a discipline by analytic method count matrix and calculated the Pearson chi-squared standardized residual for each cell (i.e., discipline and analytic method pair). The standardized residual is simply the Pearson residual (observed–expected frequency) divided by the square root of the expected frequency. The larger the positive value of the standardized residual, the greater than expected number of articles in a BLS discipline mentioning that analytic method (in its "Materials and Methods" or "Results" section). The top 10 standardized residuals for each BLS discipline are displayed in **Fig 5**.

## Analysis of analytic method groupings

For any given study, a variety of analytic methods are typically used to understand a dataset. We refer to groups of frequently co-occurring research methods as analytic method groupings, or simply method groupings. We used a data-driven tensor decomposition approach to discover commonly used method groupings across BLS disciplines. The procedure was carried out as follows: (1) analytic method by analytic method co-occurrence matrices were computed by discipline; (2) co-occurrence values within each matrix were log-transformed to reduce the influence of positively skewed analytic method counts; (3) co-occurrence matrices for each discipline were L2 normalized to remove the effect of differing total article counts between disciplines; (4) the log-normalized co-occurrence matrices were arranged into a 3-way tensor ($X^{N \cdot N \cdot D}$): analytic method ($N = 250$) by analytic method ($N = 250$) by discipline ($D = 15$); and (5) a nonnegative PARAFAC/CANDECOMP decomposition was applied to the analytic method co-occurrence by discipline tensor. The CANDECOMP decomposition factorizes a given tensor into a linear combination of R rank one tensors. In the present case, the 3-way entity co-occurrence by discipline tensor can be decomposed into a sum of R rank-one tensors, referred to as *components*, as follows (the sum of outer products of 3 vectors):

$$X = \sum_{r=1}^{R} a^r \circ b^r \circ c^r$$

For a given component, the elements of $a$ and $b$ correspond to the weights of each analytic method on the component (in the case of a symmetric co-occurrence matrix, $a = b$), and the elements of $c$ correspond to the weights of each discipline on the component. As our analytic method co-occurrence matrices represent (log-normalized) counts, we add the additional constraint that the tensor is factorized as the additive sum of nonnegative components. This has the benefit of enforcing sparsity on the components, and thus, increases the interpretability of the solution. There are no universally agreed upon criteria for the choice of R, the number of components. Analogous to some matrix factorization approaches (e.g., NMF), the more components estimated, the finer details produced in the resulting solution. However, too many components estimated may result in redundancy and/or modeling of noise. We chose a solution of 20 components, as this solution produced the most interpretable solution. Solutions with components around this number yielded similar solutions.

## Supporting information

**S1 Fig. Analytic methods trends—all categories—part 1.** Time series of 21 analytic method categories from 2009 to 2020 (annual frequency). For each analytic method category, the time series represents the proportion of articles that contained a mention of that category in their "Methods/Materials" or "Results" section per year. The time series of each analytic method category is displayed in its own plot with different y-axis scales. Note that because each plot differs in y-axis scale, caution should be observed when comparing trends across categories. Proportion of article counts by year for all analysis methods are provided in **S2 Data**. Python code for modeling trends of analysis methods is provided at https://github.com/tsb46/stats_history/blob/master/demo.ipynb.
(TIFF)

**S2 Fig. Analytic methods trends—all categories—part 2.** Time series of 11 analytic method categories from 2009 to 2020 (annual frequency). For each analytic method category, the time series represents the proportion of articles that contained a mention of that category in their "Methods/Materials" or "Results" section per year. The time series of each analytic method category is displayed in its own plot with different y-axis scales. Note that *because each plot differs in y-axis scale, caution should be observed when comparing trends across categories*. Proportion of article counts by year for all analysis methods are provided in **S2 Data**. Python code for modeling trends of analysis methods is provided at https://github.com/tsb46/stats_history/blob/master/demo.ipynb.
(TIFF)

**S3 Fig. Data-driven analysis of individual analytic method trends.** The trends in **Figs 3 and 4** display trends in superordinate categories of conceptually similar in analytic methods. For the sake of completeness, we conducted a cluster analysis of individual analytic method trends (*N* = 250) (before categorization into superordinate categories). Hierarchical clustering was performed to separate 3 clusters of analytic methods with approximately positive (Cluster 1), negative (Cluster 3), and flat (Cluster 2) trends over the study time span. **(A)** Heatmap of pairwise correlations between analytic method trends sorted according to their cluster assignment. **(B)** The trends of each cluster over the study time span (2009–2020). **(C)** The cluster assignments of all analytic methods (*N* = 250). Cluster assignments for all analysis methods are provided in **S5 Data**.
(TIFF)

**S4 Fig. Tensor decomposition weights for all 20 components.** To understand what analytic methods are frequently used together in the same study, we conducted a tensor decomposition of an analytic method co-occurrence by discipline tensor. The tensor decomposition analysis simultaneously models the co-occurrence between analytic methods, as well as their frequency of mentions in each discipline. This figure displays the discipline and analytic method weights from the tensor decomposition analysis. Components from the tensor decomposition are referred to as "method groupings" or groups of analytic methods that frequently occur together in study method and results sections. For each component, or "method grouping," a stem plot illustrates the weights for each discipline, as well as the top 10 analytic methods, in terms of their weights (sized by their weight). For each component, the discipline weights represent the frequency of usage of that component across each discipline.
(TIFF)

**S1 Data. Companion data to Fig 2.** Excel spreadsheet containing 4 tabs: (1) the total number of article counts per discipline; (2) the count of mentions for all analysis methods by year; (3)

the top 50 journal counts by discipline; and (4) the embedding coordinates of each discipline in the MDS space (Fig 2B). MDS, multidimensional scaling.
(XLSX)

**S2 Data. Companion data to Figs 3 and 4.** Excel spreadsheet containing, for all analysis methods, the proportion of articles mentioning an analysis method out of the total article count by year. The code for generating the bootstrapped confidence intervals in Fig 3 is provided at https://github.com/tsb46/stats_history.
(XLSX)

**S3 Data. Companion data to Fig 5.** Excel spreadsheet containing the standardized chi-squared residuals for each analytic method by discipline.
(XLSX)

**S4 Data. Companion data to Fig 6.** Excel spreadsheet containing results from the "method grouping" tensor decomposition analysis in 2 tabs: (1) the analysis method weights for all 20 components; and (2) the discipline weights for all 20 components.
(XLSX)

**S5 Data. Companion data to S1 Fig.** Excel spreadsheet containing the cluster assignments ($N = 3$) from the hierarchical clustering analysis of analysis method trends.
(XLSX)

## Author Contributions

**Conceptualization:** Taylor Bolt.

**Formal analysis:** Taylor Bolt.

**Investigation:** Taylor Bolt.

**Methodology:** Taylor Bolt.

**Visualization:** Taylor Bolt.

**Writing – original draft:** Taylor Bolt, Jason S. Nomi, Danilo Bzdok, Lucina Q. Uddin.

**Writing – review & editing:** Taylor Bolt, Jason S. Nomi, Danilo Bzdok, Lucina Q. Uddin.

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
