## [Editor Report · Decision Letter 0]

25 Nov 2020

Dear Dr Bolt, 

Thank you for submitting your manuscript entitled "Historical and cross-disciplinary trends in the biological and social sciences reveal an accelerating adoption of advanced analytics" for consideration as a Meta-Research Article by PLOS Biology.

Your manuscript has now been evaluated by the PLOS Biology editorial staff, as well as by an academic editor with relevant expertise, and I'm writing to let you know that we would like to send your submission out for external peer review.

Please re-submit your manuscript within two working days, i.e. by Nov 27 2020 11:59PM.

Kind regards,

Roli Roberts

Senior Editor

PLOS Biology

---

## [Decision Letter · Decision Letter 1]

9 Jan 2021

Dear Dr Bolt,

Thank you very much for submitting your manuscript "Historical and cross-disciplinary trends in the biological and social sciences reveal an accelerating adoption of advanced analytics" for consideration as a Research Article at PLOS Biology. Your manuscript has been evaluated by the PLOS Biology editors, an Academic Editor with relevant expertise, and by four independent reviewers. Please accept my apologies for the delay; we closed the journal office for two weeks over the holiday period, and have been catching up.

You’ll see that the reviewers are all broadly positive about the research question, but they each raise a number of concerns (interpretation, data presentation, search criteria and other aspects of the methodology, relationship to pedagogy). Some of these issues will need some new analyses but we believe that they should all in principle be addressable, and you should endeavour to do so for further consideration.

In light of the reviews (below), we will not be able to accept the current version of the manuscript, but we would welcome re-submission of a much-revised version that takes into account the reviewers' comments. We cannot make any decision about publication until we have seen the revised manuscript and your response to the reviewers' comments. Your revised manuscript is also likely to be sent for further evaluation by the reviewers.

We expect to receive your revised manuscript within 3 months. 

**IMPORTANT - SUBMITTING YOUR REVISION**

*Re-submission Checklist*

*Published Peer Review*

*PLOS Data Policy*

*Blot and Gel Data Policy*

Sincerely,

Roli Roberts

Senior Editor,

rroberts@plos.org,

PLOS Biology

REVIEWERS' COMMENTS:

Reviewer #1:

There are many things to enjoy in the manuscript of Bolt et al., including the scientific motivation, the approach taken, the findings, the writing, and the discussion toward graduate education.

In my view there might be few additions that could strengthen the manuscript further with seemingly little additional work.

First, to provide the frequencies of each method for every year in a table as supplemental material.

Second, to include an unbiased analysis of temporal trends. While I anticipate that the examples chosen for Figure 3 and 4 faithfully represent the main observations, they only represent individual methods. Adding an unbiased analysis (e.g.: a clustermap of all pairwise correlations of the frequencies of individual methods) with all methods could help readers to better grasp how prevalent individual temporal trends are, and which methods are more similar (the latter being something that the authors already partially address with Figure 6).

Third, to spell out the criteria used to define an entity as a method. While it is clear that the authors followed a reasonable rationale, the manuscript implies that only analytical methods relating to statistics, but not other methods such as domain-specific experimental approaches, were considered. If the latter group of methods was included in the training, its absence in the manuscript would warrant a comment on the provided data (see first point) being more extensive, and an explanation for why the manuscript only discussed one set of methods.

Other:

References to Fig 1A and Fig 1B within the text seem to point to the wrong figure.

The criteria for classifying a method into "traditional" or "advanced" could be stated out.

It is not clear why the authors chose "traditional" and "advanced". While I do not wish to impose my view on the manuscript, some clarifying words might be helpful. For instance, I would have regarded "PCA" as a "classical" method since it ultimately relates to variances and Euclidean spaces. Similarly, I would have seen "effect size" and "confidence interval" outside of "traditional" (even if not "advanced") as they would better align with non-parametric tests and data that does not follow a narrow set of characteristics that would allow the appropriate usage of t-tests or ANOVA etc.; Similarly, I would anticipate "effect size" and "confidence interval" to align with "big" data, where the size of the dataset would avoid the need to use some other "traditional" approaches to estimate the spread of the data. 

Possibly already addressed by my first main point, I would be curious on the temporal trends of those special cases of methods that can be used to evaluate whether other methods are suitable (e.g.: Lilliefors). 

Providing the accuracy of the classification of disciplines on a per-discipline level, rather than only in aggregate, could assist people from smaller disciplines to better asses the strength of the findings of this manuscript within their own discipline. 

It is unclear how the classification of disciplines would be affected by articles that do not fall into any of the 15 chosen classes - or if this could be a problem that could likely be safely ignored (e.g.: if only applicable to few publications).

Reviewer #2:

Review report for "Historical and cross-disciplinary trends in the biological and social sciences reveal an accelerating adoption of advanced analytics" by Bolt et al.

This paper explores the cross-disciplinary usage of different research methods by analyzing a large corpus of research publications largely in biological and social science fields. Specifically, the authors leveraged data-driven text-mining tools to identify research methods from publication texts and then analyzed the cross-disciplinary trends over the past 10 years. The authors reported several major findings: (1) The classical statistics and regression-based methods have declined in the past decade, while machine-learning approaches have seen a significant increase. (2) Research method categories are uniquely associated with each research discipline. (3) There are co-occurrence patterns in research method usage, where research groupings can be further decomposed into cross-discipline and within-discipline method categories. Lastly, the authors suggested some policy implications of their findings, for example, on cross-disciplinary collaboration, training of researchers, and education in research methods. 

The paper considers an interesting research question and reveals an important trend in research disciplines, showing the ongoing methodological shifts in a variety of scientific communities. In particular, as the development of novel research methods, computational tools, and large-scale datasets, there may be substantial shifts in the methodology and framework of doing research in many traditional fields. For example, the advances of deep learning in recent years have not only promoted the rapid development of computer science fields but also remarkably benefited other research fields in prediction and computation. Here, the Alpha Fold can be a nice example (Improved protein structure prediction using potentials from deep learning. Nature 577, no. 7792 (2020): 706-710). In this sense, I think this paper has its merit in providing some initial evidence on the adoption of advanced analytics across several fields. However, in my view, this paper still has room to improve before its consideration for publication. In the following, I would like to provide some comments and suggestions, which may be helpful for the authors to further strengthen their paper. 

General comments:

The authors observed that the usage of some traditional research methods (such as t-test, ANOVA, and regressions) has declined in the publications over the considered period, while the advanced analytics (such as machine learning methods) have seen an increase during the same period. The results are seeming obvious and the interpretation of these major observations remain largely limited in this paper. First, what are the driving forces behind these observations? These could be due to the publication landscape shifts towards computing and statistical fields, the changes of research tasks of traditional fields, and the availability of new types of data sources. For example, the use of LandScan satellite image data in the study of social and economic status inference requires image processing methods other than traditional regression methods based on survey and census data [Combining satellite imagery and machine learning to predict poverty. Science 353, no. 6301 (2016): 790-794]. Second, what was the baseline to understand the relatively share or the adoption of research methods across different methods? By definition, some research methods and tools are old and traditional, and some others are new and advanced. Even for new and advanced general-purpose methods/technologies, it takes some time for them to be used by many fields. Also, they will be replaced by more recent ones as technology advances. It is not clear enough how to compare the observed patterns with a baseline. Third, related to the first point, advanced analytics have been increasingly used, but it is not clear if they are used to address the same research tasks or new research tasks in one research field. In my view, without these further explorations and in-depth understanding of these observations, the contributions of this paper are best limited.

This paper relied on the entity recognition of research methods based on the description of Methods and Materials (method, material, measure, analysis, and statistical) in the publication texts. The authors first developed an initial list of 700 phrases that are commonly used research methods. Then, they applied a statistical named entity recognition algorithm to the entire corpus of methodology section texts to generate the final list of detected research method entities. Finally, they pre-processed the list of method entities and resulted in 1,218 final method entity vocabulary. There are several potential issues on the method. First, not all papers have the specific section of Methods and Materials. For many journals in social and economic journals, the method, material, measure, analysis, and statistical sections are also not standardized. It is not clear how the selection of the body of full text affect the results. Also, the mention of research method entity does not mean the use of them. For example, the Methods section of this paper (which includes many subsections, pages 13-18) mentioned a lot of research method entities, however, not all of them are used in this paper (If the authors apply their method to this paper, what can be extracted and identified?). The use of full text of Methods and Materials provides more information, while, at the same time, this may also introduce as much noises. Second, the recognition of research method entity is lack of independent validations. The used statistical named entity recognition algorithm and the convolutional neural network under the SpaCy package seem to be a black box. The 16,020 research method entities discovered by the algorithms could be very messy, and the resulting 1,218 final method entities seem to be only a very small portion (less than 10%). The used method and the list are lack of transparency. It remains unclear how the selection of entities affects the results, how to validate the recognized entries, and whether they are meaningful. Third, the author used data from Pubmed database, which covers mainly biological and social science fields. Meanwhile, the authors only included a random subset of Plos One articles in the corpus. It remains unclear how the sampling strategies affect the results. For example, whether the results are robust if using the full sample of publications such as Microsoft Academic Graph (MAG) data and Web of Science (WoS) data. Fourth, the authors used a machine-learning text classification approach to classify each single article into a set of disciplines. The method is not clearly presented, and it remains unclear whether the identified fields are well aligned with the MAG fields or WoS fields. Additional independent validations are necessary. 

Specific comments:

1. In the abstract section, the authors presented "At the same time, there is increasing concern that education in quantitative methods is failing to adequately prepare students for contemporary research. These trends have led to calls for educational reform to undergraduate and graduate quantitative research method curricula". However, I failed to find evidence or references in the introduction section that can support these claims. In my view, these sentences would be better to be placed in the discussion section or relocated to (after a revision) the end of the abstract as the implications of this paper. 

2. In the introduction section, it would be better to add some discussions on the motivation of this paper. In particular, it would be nice if the author can develop some hypotheses in this section and then test them using data. In other words, what would the readers anticipate after reading the introduction could be better covered. In addition, some references can be used to better support the authors' narrative, for example, the open science in the past decade [Foster, E. D., & Deardorff, A. (2017). Open science framework (OSF). Journal of the Medical Library Association: JMLA, 105(2), 203], and the use of large-scale data and interdisciplinary methods in social and economic fields [Computational socioeconomics. Physics Reports 817 (2019): 1-104]. Moreover, the core results of this paper could be embedded in the introduction section. The current introduction was too brief in my view. 

3. In the pre-processing workflow of research method entity, the granularity of the research entity should be justified. It seems that "statistically significant" is the same as "statistical significance", but they are two method entities in this paper (on page 4). Some methods are essentially the same but named differently, for example, Gaussian Process regression and kriging in geostatistics. How will these issues affect the results? In addition, the 15 research disciplines of publications should be cross-validated. The relationship between the classification of research fields and the grouping of method entities is not clear. It seems not necessary that "the pre-processed method entities before the entity disambiguation step were supplied to a third analytic pipeline". In addition, one paper can still belong to multiple research disciplines, correct? If one paper used Method A for 10 times and Method B for 1 time in a paper, does the entity recognition algorithm count them once each or different times?

4. On page 4 of this paper, the "Figure 1A" and "Figure 1B" should all be "Figure 2A" and "Figure 2B". Moreover, it is not clear how to interpret Figure 2B. The size of nodes shows the Discipline Article Counts, but they are not in the correct size. The authors presented that "This approach made apparent that four disciplines stood out as relative outliers in research method usage: Oncology, Population Genetics, Evolution/Ecology and Chemistry/Material Sciences. As we observe below, among all 15 candidates, these select disciplines revealed a unique profile of research method usage". However, from Figure 2B, I could not find these observations, for example, there seems no evidence showing "four disciplines stood out as relative outliers". In the caption of Figure 2, the author presented "The research disciplines with the highest article counts were primarily biomedical and clinical disciplines". Yet, the evidence is also missing. 

5. At the top of page 6, the authors presented that "Interestingly, the only classical statistical methods that we observed to gain in adoption over the past decade are effect sizes and confidence intervals. The increasing adoption of effect sizes and confidence intervals may reflect the increased pressure from institutions and researchers (10-12) to report effect sizes and confidence intervals along with significance tests in peer-reviewed research". In my view, the reason behind the increasing adoption of effect sizes remains unclear. I am wondering if the authors could provide some evidence to support these claims.

6. In Figure 3, the authors used different scales in the y-axis for research disciplinary categories. Yet, it is very hard to make the comparison across different categories. It would be also nice to show the figure that used 0 as the starting point for each panel, for example, in the online supplementary information. Moreover, there could be field differences, for example, the uses of neural networks are different in the computer science field and in the social science field. It would be interesting to explore more and include these new results in the supplementary information. 

7. In the section "Unique Method Usage by Discipline", the authors identified some method categories uniquely associated with each discipline. The word "unique" here is misleading because the same research method entity can still be present in multiple research disciplines. The authors used a contingency table approach (the Pearson chi-square standardized residual for each cell) to identify the associated the category. I am wondering if the approach is the same as the "revealed comparative advantage" [Trade liberalisation and "revealed" comparative advantage" The Manchester school 33, no. 2 (1965): 99-123], which is widely used in many fields such as [The product space conditions the development of nations." Science 317, no. 5837 (2007): 482-487]. Not sure if they are similar or the same but with different names.

8. In Figure 6, the authors only presented part of all 16 components. It would be better to present all of them, otherwise the reason of excluding some of them should be given. In the caption, the authors presented that "a stem plot illustrates the weights for each discipline, as well as the top 5 method entity strings". However, it seems that top 10 method entities were given, not top 5. Moreover, the author presented "Some sets of methods are represented across all BLS disciplines (e.g., component 2), while others are concentrated within one or two disciplines (e.g., component 15)". Yet, this observation is very tricky because the scale of the x-axis (discipline weight) is very different. The authors are suggested to use the same scale for the x-axis. 

9. In the Methods section (and the abstract section), the sample size is no longer a total of 3.5 million articles. Only 1,276,452 articles have the sections of Methods and Materials. I am wondering if this should be the starting point of the sample size. Moreover, it is not clear why the authors thought the PLOS ONE data is biased. It should not because it contained a disproportionate number of articles in the corpus. If so, all journal data is biased because they did not publish the same number of papers. In addition, the total number of method categories resulting from our manual classification was 126. It is not clear the granularity of the classification and the quality of the grouping. 

10. It seems that a majority of the Methods section can be relegated to the online supplementary information. In the reference section, some journals are abbreviated while others are not. The authors are suggested to use the same standard. 

Reviewer #3:

Overall, the study addresses a very interesting and a significant practical issue concerning the use of research methods. Text mining techniques are applied to the methods sections in a collection of open access publications in biomedicine and some other disciplines.

PLOS Biology may be a good venue to publish the work, but I am wondering whether PLOS One would be more appropriate, given that the work would be accessible to other disciplines as well as biology.

The organization of the presentation is not straightforward. It should follow the logical flow, i.e., your question, your method, results, discussions and conclusions. Currently, it reads like a montage of various descriptions.

What is commonly missing from the presentation is some minimal justifications of the choices you made, especially when multiple options are apparently available and even some of better alternatives are available as well. A large number of analytic techniques are used, MDS, tensor decomposition, CNN, Eucliedian-based similarity metric, etc. Is the wide variety of tools strengthening the study overall as opposed to using a small number of tools but each one digs deeper instead?

Some relatively minor issues are identified below:

The title is too complex and too long. Something like the short title would be much clearer.

Since the initial ~3.5 million was narrowed down to a subset and the methods section was the target, you should just mention the number of sections that satisfied your selection criteria.

Does the scope of BLS accurately reflect the study? It is not clear from the introduction.

'computer- or qualitative-based assessment of measured data points'

how about:

quantitative and qualitative assessments of data.

The Introduction section is immediatedly followed by the Results section. Since many concepts and operations are not yet explained, it is too soon to present the results.

What is described in the Results section is not really about results. Instead, they are the steps you take. More details

are necessary here. For example, "We used a named entity recognition algorithm trained specifically for this purpose."

What exactly does the algorithm do? What are the specific requirements to meet for this specific purpose?

Similarly, a supervised machine learning framework was used. More details and justifications should be provided. To create

15 research disciplines, you extracted information from titles, abstracts and journal names, why not from keywords,

which should be available from many of the articles and they are likely to improve the quality of your discipline identification and classification.

Figure 1: step 4 is not numbered in the pipeline. Perhaps you should include an example of how article-level information is

used for article classification.

The first mention of Figure 1A and 1B, do you mean Figure 2A and 2B?

"We deployed classical multidimensional scaling (MDS) ... (Figure 1B). You should give a brief justification of your

choice of using MDS. There are a few similar issues with the use of other methods and tools without a proper justification,

especially when multiple options are available, for example, tensor decomposition.

Plots in Figure 3 are not on the same scale, although they all represent proportion, or percentage. For example, the Summary

Statistics on the upper left ranges from 0 to 0.60, whereas Analysis of Variance (ANOVA) next to its right ranges from 0

to 0.125.

Figure 5 is very interesting. Methods at various levels of scope and abstraction are presented together, e.g., meta

analysis and Z Score normalization.

The Discussion section

'Either way, our finds reinforce the concern that statistics and research method education in the BLS science is falling

behind ...' It is not clear to me how you can draw this conclusion from your results since the education was never

part of the study.

goals of select research methods -> goals of selected research methods

'The appropriate context for each test is controversial among statisticians (33, 34).' It would be informative if you can

give us more information about the controversy so we can see the connection between the controversy and any patterns shown

in your results.

Methods

'To ensure ... not biased ..., we selected 2,625 random ...texts with tagged phrases ... and fed them to ... NER...' I

can see you may capture those that are similar to the 700 phrases with this process, but it is not clear how going

through this process will eliminate the bias if the 700 phrases are biased to begin with.

Discipline Similarity Analysis

The Euclidean distance: why did you choose this particular metric in this case? Is this the most appropriate metric to

use in this case?

Reviewer #4:

In this meta-research article, the authors propose that examining the frequency with which different statistical methods are used in the literature will allow scientists to tailor education in data analysis to the needs of students in particular field. They assess the statistical methods used in a large body of open access papers and conclude that use of conventional statistical methods is decreasing, while use of developing data science techniques is increasing. They further examine field specific patterns. While the idea is compelling, the authors consistently rely on small differences over time or between disciplines when interpreting their data, without consideration of baseline values or the magnitude of observed changes/differences. This creates the impression that they are consistently making choices when presenting the data to show that more training in data science is needed, rather than interpreting the data as a whole in an unbiased manner. Specific concerns are as follows:

1. Figure 1 is quite difficult to interpret. In panel A, the abbreviations are not always intuitive, forcing the reader to switch back and forth between the figure and legend so frequently that the message is lost. The small font in the figure exacerbates this. The figure is set up to show the proportion of papers in each journal that fall with in specific fields; yet many readers are likely more interested in what journals are represented in fields that are of interest to them. In panel B, there is no explanation of what the two dimensions represent. The legend on circle size isn't tied to specific numbers of articles, and the circles in the legend are generally much larger than those in the graphs. The legend provides no information that would help one to interpret panel B. In panel C, there is again no legend tying specific rows to entity counts. Readers are supposed to assess differences in counts based on font size, however this is not something that the human eye can accurately do. Groupings based on conceptual similarities between different entities are discussed in the text, but do not appear in the figure. This figure needs to be redesigned to convey a clear message.

2. The authors choose the term "method entities" to refer to statistical analysis methods throughout the paper. Given that their focus is on analytic methods, a more specific term may be beneficial. Many readers will assume that method entities refer to the complete range of laboratory and clinical methods used in their field.

3. Figures 3, 4: The legend of figure 3 suggests that the top panel is missing. Both figures use small multiples; yet the authors have violated a key rule of small multiples by using completely different scales in each graph. This makes it impossible to tell which methods are common vs. uncommon, and which increases/decreases are small vs. large. The authors have done this because some of the percentages and increases are so small that they would not be visible on a uniform scale. One has to look closely to see that many of the machine learning categories, for example, are used in less than 1% of papers. In contrast, the traditional method categories are generally used in 10-15% of papers. These observations illustrate the danger of drawing conclusions based solely on increasing or decreasing patterns, without reference to baseline values or the magnitude of the observed changes.

4. Similar concerns apply to figure 5 - the data presented focuses on differences between observed and expected, without consideration of how frequently each method is used within the given fields. The authors' hypothesis is that examining the frequency with which methods are used, and patterns of change, will allow us to tailor education within different disciplines. Yet, readers cannot use this figure for this purpose, as it doesn't consider the frequency with which methods are used. A data driven summary figure showing commonly used methods in each field, in addition to emerging techniques in which students would benefit from additional training, may be beneficial.

5. Validation and performance characteristics for the different methods used is lacking from the material and methods section. The authors describe how things were done, but they don't specify how well the algorithms work compared to gold standard ratings.

6. While the article includes a limitations section, it is limited in scope. Authors may be more likely to report details of new or complex methods, compared to commonly used methods like t-tests and ANOVA, which would lead to undercounting and bias. Statistical reporting is poor in many fields. The authors may be able to incorporate checks into their data to assess the frequency of completely inadequate reporting (i.e. p-values reported, but no methods that generate p-values are listed). 

7. The authors approach examines the frequency with which different methods are used; it cannot assess whether authors are using the appropriate method for the data that they are analyzing. It is likely that there is additional need for newer methods in many fields; yet uptake is limited by a lack of expertise. Differences in expertise and training for new methods across fields may also contribute to limited uptake in some fields compared to others, magnifying the field specific differences that the authors observe. Using data like those generated by the authors to set educational priorities could potentially exacerbate these problems. While data driven approaches are beneficial, one should not lose site of the fact that some of the excitement around these approaches is based on the potential of what could be achieved if they were well implemented on a broader scale. It may be too early to capture this using the approach described by the authors. Furthermore, a clear discussion of what poor vs. good implementation looks like is needed - a proliferation of poorly conducted studies using the latest popular methodology has hindered progress in many fields.

---

## [Decision Letter · Decision Letter 2]

19 May 2021

Dear Dr Bolt,

Thank you for submitting your revised Meta-Research Article entitled "The Changing Landscape of Analysis Methods in the Biological and Social Sciences" for publication in PLOS Biology. I have now obtained advice from three of the original reviewers and have discussed their comments with the Academic Editor. 

Based on the reviews, we will probably accept this manuscript for publication, provided you satisfactorily address the remaining points raised by the reviewers. Please also make sure to address the following data and other policy-related requests.

IMPORTANT:

a) Please address the remaining requests from the reviewers, including the additional robustness checks for rev#3 and the presentational requests from rev #4.

b) The current title is fine, but it might be more appealing to readers if it mentions (in a few words) the implications of your findings, e.g. for education of future generations of biologists.

c) Please address my Data Policy requests below; specifically, please supply numerical values underlying Figs 2ABC, 3, 4, 5, 6, S1AB, S2AB, S3, and cite the location of the data clearly in each relevant Fig legend.

d) Please supply a blurb, following the instructions in the submission form.

e) Your financial declaration currently says that you had no specific funding for this work; can you just confirm that this is correct, or add appropriate funding information?

We expect to receive your revised manuscript within two weeks. 

*Published Peer Review History*

*Early Version*

Sincerely,

Roli Roberts

Senior Editor,

rroberts@plos.org,

PLOS Biology

DATA POLICY:

Regardless of the method selected, please ensure that you provide the individual numerical values that underlie the summary data displayed in the following figure panels as they are essential for readers to assess your analysis and to reproduce it: Figs 2ABC, 3, 4, 5, 6, S1AB, S2AB, S3. NOTE: the numerical data provided should include all replicates AND the way in which the plotted mean and errors were derived (it should not present only the mean/average values).

DATA NOT SHOWN?

REVIEWERS' COMMENTS:

Reviewer #1:

The authors have addressed all my points.

As the new figure with the clustermap is referred to as Supplemental Figure 2 within their reply, but as Supplemental Figure 1 within the manuscript, they might verify that the placing of this Supplemental Figure within the manuscript is as intended.

Reviewer #2:

Review report for "The Changing Landscape of Analysis Methods in the Biological and Social Sciences" by Bolt et al.

I would like to thank the authors for considering my comments and suggestions on the previous version of the paper. In my view, the authors have tried to improve their work and they did a nice job. I am largely satisfied with their responses to my comments. In this round, I only have a few minor comments which may be helpful for the authors to further strength their paper.

1. The authors argued that the number of references to the same analytic method in a paper does not provide evidence for the frequency of its usage in the paper. Also, it is possible that some entities identified by the NER algorithm were not actually used in the paper. I am wondering what the results would look like if the authors do a weighted version of entities, i.e., the intensity of references to an analytic method by a research discipline. Maybe include some robustness checks in the supplementary.

2. The authors acknowledged that one paper can belong to multiple research disciplines, for simplicity here each paper was categorized into the dominant discipline. They have included this simplification as a limitation in the discussion section. Still, I am wondering if the multi-classification of papers would have a significant impact in the results considering the increasing trend of interdisciplinary papers especially in recent years. Maybe include some robustness checks in the supplementary. 

3. Besides the mention of a single method by different disciplines, it may be also interesting to consider the co-occurrence of two method entities across disciplines, or a group of entries that are co-occurred very often. In my view, this is another promising direction for future studies, and this can be mentioned in the discussion section.

4. I found the first few sentences of the abstract and the first paragraph of the introduction particularly interesting, considering that the changing use of methods across disciplines may lead to educational reform to undergraduate and graduate quantitative research method curricula. It may be good to echo these broad conservations in the discussion sections and suggest more policy implications of the findings in this paper. 

5. Some minor suggestions on figures:

(a) The x-axis in Figure 2A can be round to "int". 

(b) The legend of Figure 3 can be sorted by the result of the latest year 2020. This alignment will better guide the eyes of readers. 

(c) The resolution of Figure 4 should be largely enhanced. In addition, the legends in some panels are much overlapped with the curves. May consider a two-column legend layout when it helps. 

Overall, the authors did a great job. Good luck with the submission and publication. 

Reviewer #4:

Fig 3 - The legend has many colors; it will be difficult to make the color palette accessible to colorblind readers, and even readers with normal color vision will need to search through and color match to find what they are looking for. To enhance readability, the authors might consider labeling each line on the right hand side (where the legend is) next to the line. This would allow readers to quickly scan down and determine which methods are most common. Colored lines from the legends could still be used for cases when different lines are close together.

Fig 4 - The authors may want to consider providing a "y-axis location" legend; a small box or rectangle next to the legend of each figure sized so that the y-axis runs from the minimum to the maximum of any panel shown (0 to 0.65), on which the region encompassing the scale for the y-axis of the particular graph is highlighted. This would give readers a quick visual indicator to help with comparing across graphs. (For example, for NHST, the region from 0 to 0.25 would not be highlighted, whereas the region from 0.25 to 0.65 would be highlighted. For PLS/DA, the region from 0 to 0.02 would be highlighted).

---

## [Editor Report · Decision Letter 3]

7 Jun 2021

Dear Dr Bolt,

On behalf of my colleagues and the Academic Editor, Ulrich Dirnagl, I'm pleased to say that we can in principle offer to publish your Meta-Research Article "Educating the Future Generation of Researchers: A Cross-Disciplinary Survey of Trends in Analysis Methods" in PLOS Biology, provided you address any remaining formatting and reporting issues. These will be detailed in an email that will follow this letter and that you will usually receive within 2-3 business days, during which time no action is required from you. Please note that we will not be able to formally accept your manuscript and schedule it for publication until you have made the required changes.

PRESS: We frequently collaborate with press offices. If your institution or institutions have a press office, please notify them about your upcoming paper at this point, to enable them to help maximise its impact. If the press office is planning to promote your findings, we would be grateful if they could coordinate with biologypress@plos.org. If you have not yet opted out of the early version process, we ask that you notify us immediately of any press plans so that we may do so on your behalf.

Thank you again for supporting Open Access publishing. We look forward to publishing your paper in PLOS Biology. 

Sincerely,

Roli Roberts

Roland G Roberts, PhD 

Senior Editor 

PLOS Biology